# Vertical organic synapse expandable to 3D crossbar array

Yongsuk Choi[1,4], Seyong Oh[2,4], Chuan Qian [1], Jin-Hong Park [2,3✉] & Jeong Ho Cho[1✉]

Recently, three-terminal synaptic devices have attracted considerable attention owing to their nondestructive weight-update behavior, which is attributed to the completely separated terminals for reading and writing. However, the structural limitations of these devices, such as a low array density and complex line design, are predicted to result in low processing speeds and high energy consumption of the entire system. Here, we propose a vertical three-terminal synapse featuring a remote weight update via ion gel, which is also extendable to a crossbar array structure. This synaptic device exhibits excellent synaptic characteristics, which are achieved via precise control of ion penetration onto the vertical channel through the weight-control terminal. Especially, the applicability of the developed vertical organic synapse array to neuromorphic computing is demonstrated using a simple crossbar synapse array. The proposed synaptic device technology is expected to be an important steppingstone to the development of high-performance and high-density neural networks.

[1] Department of Chemical and Biomolecular Engineering, Yonsei University, Seoul 120-749, Republic of Korea. [2] Department of Electrical and Computer Engineering, Sungkyunkwan University, Suwon 16419, Republic of Korea. [3] SKKU Advanced Institute of Nanotechnology (SAINT), Sungkyunkwan University, Suwon 16419, Republic of Korea. [4]These authors contributed equally: Yongsuk Choi, Seyong Oh. ✉email: jhpark9@skku.edu; jhcho94@yonsei.ac.kr

With the rise of the "big data" era, in which there has been an explosion of unstructured data, such as images, text, sound, and video, handling such types of data through using conventional von Neumann computing with separate processing and memory units has become difficult[1–5]. Neuromorphic computing–which mimics the ability of the human brain to perform energy-efficient parallel processing of information using a complex neural network (NN)–has attracted considerable attention as one of the pathways to meet such technical demands[6–10]. The brain processes and memorizes information simultaneously, which makes it free from the bottleneck problem. As such NNs in the brain consist of numerous synapses, the development of high-density and low-power synapse-like devices is essential to the successful implementation of neuromorphic computing[1,2,4,11–14]. As pioneering research, extensive studies on an artificial synapse based on a two-terminal resistive memory device have been conducted in recent years[4,7,11,13,15–17]. These two-terminal synapses are fabricated in a crossbar array structure, whose simplicity and short channel ensure a high integration density and low power consumption. However, nondestructive weight update in the two-terminal synapse is difficult to be accomplished because of its structural nature, i.e., a single shared terminal for reading and writing[7,15–21]. Very recently, Wang et al. effectively alleviated this issue by applying a significantly low readout voltage pulse, but further researches are still required for resolving this issue fundamentally[17]. In the meantime, three-terminal synaptic devices have attracted considerable interest owing to their nondestructive-weight-update behavior, which is attributed to the completely separated terminals for reading (drain) and writing (gate)[1,6,9,22–27]. In recent studies, three-terminal artificial synapses implemented with various inorganic and organic materials showed a desirable weight-controllability property via various charge-storage principles using interfacial traps[28–30], atomic vacancies[14], ion intercalation[22,26,28,31], and floating gates[32–35]. For example, electric-double-layer transistors and floating-gate transistors have been demonstrated to be able to successfully emulate a biological synapse[12,32,33,36]. However, three-terminal synapses have a lower array density and a structural limitation on line-design compared to the two-terminal crossbar array structure in a complicated circuit configuration; these drawbacks result in a lower processing speed and higher energy consumption of the entire system.

Herein we propose a vertical synapse featuring a remote weight update via ion gel, which is also extendable to a crossbar array structure. For the device configuration, a sub-100-nm-thick poly (3-hexylthiophene) (P3HT) channel is positioned at every cross-point of the pre- and postsynaptic terminals, and the ion-gel weight-control (WC) layer is deposited on them. Mobile ions in the ion gel readily penetrate the free volume in the P3HT channel, which results in a nonvolatile change in the channel conductance. Important synaptic properties, such as short-term plasticity (STP), excitatory and inhibitory postsynaptic currents (EPSC and IPSC, respectively), and long-term potentiation/depression (LTP/D) are evaluated via current–voltage measurements. In particular, the dimensional condition of vertical channel for achieving the optimal LTP/D characteristics are investigated via control of the channel length and area of the line cross-point. Finally, the applicability of the developed organic synapse array to the hardware NNs (HW-NNs) is evaluated in two ways: small-scale real-time learning and large-scale theoretical simulation.

## Results

**Fabrication of ion-gel-gated vertical crossbar synapse array and its synaptic properties**. Figure 1a, b show a proof-of-concept

illustration of vertical crossbar synapses with two-terminal and three-terminal device geometries, respectively. The three-terminal vertical synapses, enabled by remote gate controllability of ion-gel, are integrated into a large area crossbar array to construct an artificial neural network (ANN) as shown in Fig. 2a. The device array consisted of the bottom and top electrode lines, which corresponded to the pre- and postsynaptic terminals, respectively. The semiconducting polymer layer, sandwiched at every cross-point between the top and bottom electrode lines, served as the synaptic channel. The ion-gel layer and top gate lines were utilized to achieve the nonvolatile-weight-change property of a biological synapse as a WC terminal stack. By virtue of the free volume in the semiconducting polymer layer, mobile negative ions in the ion-gel could readily penetrate the channel under the application of a negative WC voltage ($V_{WC}$)[37,38]. The penetration of negative ions (TFSI$^-$ ions) was proven using scanning electron microscope-energy dispersive X-ray spectroscopy (SEM-EDS) analysis (see details in Supplementary Fig. 1 and Supplementary Note 1). In contrast, ions moved out from the channel layer under the application of a positive $V_{WC}$. The penetrating negative ions accumulated hole carriers in the channel layer, which led to an increase in the channel conductance. The conductance change in our ion-gel-gated device caused by the ion movement was similar to the operation of a biological synapse[39–41]. In the proposed ion-gel-gated artificial synapse, p-type P3HT was used as the vertical channel, whereas an ion-gel consisting of ionic liquid and poly(vinylidene fluoride-co-hexafluoropropylene) (PVdF-HFP) was used as the gate dielectric layer (see the chemical structures in Fig. 2a). Figure 2b shows a cross-sectional schematic of the fabrication procedure of the proposed ion-gel-gated vertical synapse. To fabricate the organic synapse array, a P3HT solution blended with a crosslinking agent was spin-coated onto the substrate with prepatterned bottom metal lines[42,43]. The channel was then defined by UV exposure and a subsequent solvent washing process. The top metal line was thermally deposited to form the vertical channel. Finally, the ion-gel gate dielectric layer was spin-coated, and the WC electrode lines were thermally deposited. Figure 2c shows an optical-microscopy image of the vertical transistor-type organic synapse array.

The channel conductance of a transistor-type synaptic device is correlated to the synaptic weight in a biological synapse[17,24]. Thus, we applied a $V_{WC}$ pulse to the WC terminal and monitored the postsynaptic current (PSC) between the bottom (source) and top (drain) lines. A typical PSC–$V_{WC}$ characteristic curve of the ion-gel-gated vertical synaptic device is shown in Supplementary Fig. 2. The device showed a high on/off ratio of over $10^5$ within a small $V_{WC}$ operation range of ±4 V. A clear hysteresis loop was observed in the clockwise direction because of the slow movement of the penetrating ions within the P3HT channel[37,38]. Then, the synaptic properties, including the EPSC/IPSC, paired-pulse facilitation (PPF), and LTP/D, of the ion-gel-gated vertical synapse were analyzed. Under varied $V_{WC}$ with its amplitude from ±0.5 to ±3 V and width of 50 ms, the PSC measured at a constant presynaptic voltage ($V_{pre}$) of −0.01 V showed clear EPSC and IPSC responses (Fig. 2d). The PSC immediately increased (decreased) upon the application of negative (positive) $V_{WC}$, and it was retained even after 50 s; however, it did not return to the initial value, because of the residual ions in the P3HT layer.

Next, the STP of our synaptic device was investigated. Figure 2e shows the PPF characteristics of the device measured under a small $V_{WC}$ of −1 V. In this voltage, ion penetration into the P3HT channel was limited, and thus, the device showed clear STP behavior. As shown in the inset of Fig. 2e, two successive pulses applied at a shorter interval evoked highly amplified EPSC responses. The PPF index can be defined as the ratio of the first

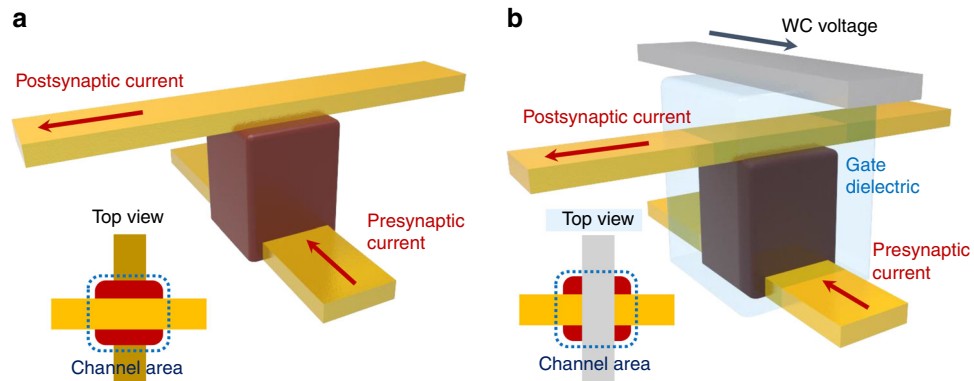

**Fig. 1 Schematic illustration of synaptic devices. a** Two-terminal synapse. **b** Three-terminal synapse with crossbar array structure.

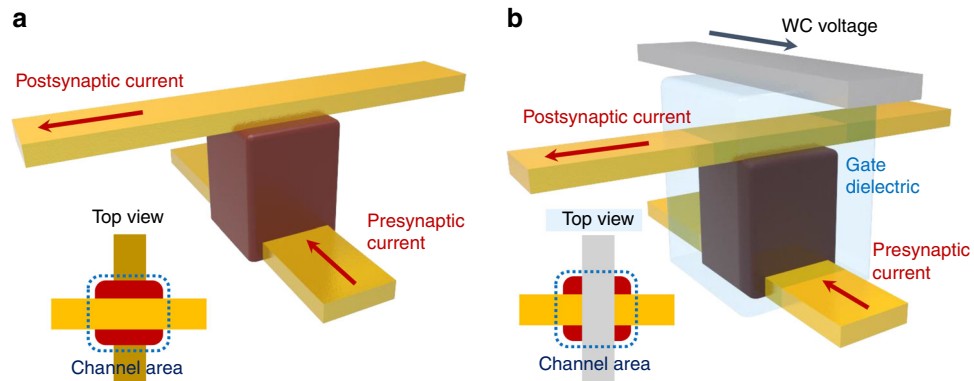

**Fig. 2 Fabrication of ion-gel-gated vertical crossbar synapse array and its synaptic properties. a** Schematic diagram of ion-gel-gated vertical crossbar synapse array mimicking biological NN. The inset shows the chemical structures of P3HT, PVdF-HFP, and ionic liquid ([TFSI]⁻ anion and [EMIM]⁺ cation). **b** Cross-sectional schematic of fabrication procedure of ion-gel-gated vertical P3HT synapse. **c** Optical microscopy image of crossbar synapse array. **d** EPSC and IPSC responses induced by negative and positive $V_{WC}$s with magnitudes varying from ± 0.5 to ± 3 V. **e** Plot of PPF index as a function of $\Delta t$ for proposed ion-gel-gated P3HT crossbar synaptic device. The inset shows the PSC generated by a pair of $V_{WC}$ stimuli. **f** LTP/D characteristics of synaptic device under application of 100 consecutive potentiation pulses ($V_{WC} = -3$ V) followed by 100 consecutive depression pulses ($V_{WC} = +2$ V).

PSC peak to the second PSC peak $(A_2/A_1)$[39,41]. This PPF index decreased exponentially as the interval between the first and second pulses ($\Delta t$) increased. The relationship between the PPF index and the interval can be expressed as PPF = 1 + $C_1$ exp $(-\Delta t/\tau_1)$ + $C_2$ exp($-\Delta t/\tau_2$), where $\tau_1$ and $\tau_2$ denote the relaxation times of rapid and slow phases, respectively, and $C_1$ and $C_2$ are constants representing the initial facilitation magnitude of rapid and slow pulses, respectively. Our ion-gel-gated vertical synapse showed $\tau_1$ of 87 ms and $\tau_2$ of 1762 ms, which coincided with the corresponding values of a biological synapse[39–41]. Our synaptic device also showed reliable long-term plasticity behavior wherein the changed current did not completely return to the initial value, because of the residual ions in the P3HT channel. To analyze the LTP/D characteristics of the proposed device, 100 consecutive potentiation pulses ($V_{WC} = -3$ V, 50 ms) followed by 100 consecutive depression pulses ($V_{WC} = +2$ V, 50 ms) were applied to the WC terminal. As shown in Fig. 2f, the PSC after 100 negative pulses increased up to 15.3 mS, after which it decreased continuously with the application of positive pulses. Overall, the device successfully mimicked various electrical behaviors that occur in biological synapses, including short-term and long-term properties.

**Optimization of vertical synaptic device geometry.** The LTP/D characteristics of a synaptic device are considered as its most critical property in neuromorphic computing because key factors of the LTP/D curves, such as the dynamic range ($G_{max}/G_{min}$), nonlinearity (NL), and effective number of states ($NS_{eff}$), have a significant impact on the accuracy of the learning/recognition tasks performed by an ANN[44,45]. Therefore, we optimized the LTP/D characteristics of the proposed device by adjusting the ion-penetration behavior through tuning of the channel thickness and area of the ion-gel-gated vertical synapse. Figure 3a and Supplementary Fig. 3 show the LTP/D characteristics of vertical synaptic devices with P3HT channels of various thicknesses

under the application of a set of $V_{WC}$ pulses consisting of 100 potentiation pulses ($V_{WC} = -3$ V) and 100 depression pulses ($V_{WC} = +2$ V). Here, the pulse frequency and pulse width were fixed at 2 Hz and 50 ms, respectively (see Supplementary Figs. 4 and 5, and Supplementary Notes 2 and 3 for the additional information about various pulse frequencies and pulse widths). The thickness of the P3HT channel was controlled in the range of 20–95 nm by varying the concentration of the P3HT solution, whereas the channel area was fixed at $50 \times 50$ μm². The maximum conductance value ($G_{max}$) was the highest (18.9 mS) in the device with the thinnest channel (20 nm), and it decreased to 2.3 μS with an increase in the thickness of the P3HT channel to 95 nm. Because the channel thickness is considered as the channel length in a vertically stacked device, in this study, the channel length of the synaptic device increased with increasing thickness of the P3HT channel; this resulted in a decrease in the overall channel conductance. The dynamic range ($G_{max}/G_{min}$) values of the synaptic devices having 20-, 35-, and 55-nm-thick P3HT channels were higher than 10, which is the minimum value required for a successful pattern recognition task[45]. To extract the values of NL and $NS_{eff}$, which represent the precision of the weight-update behavior, we first normalized the LTP/D characteristic curves of the devices with different channel lengths by dividing each conductance value by the maximum value ($G/G_{max}$), as shown in Fig. 3b. Then, we calculated the NL value by fitting the measured curve to the normalized one (see detailed equations in the methods section, Supplementary Figure 6, and Supplementary Note 4). The synaptic device with the thinnest channel (20 nm) showed a positive NL value (+5.4). In this device, the ions were able to penetrate the entire channel upon the application of $V_{WC}$ pulses, which enhanced the overall channel controllability. In contrast, the NL values of the devices with the thicker channels decreased in the negative direction and reached −0.6 for the device with the 55-nm-thick P3HT channel. This is because the channel region far away from the ion gel was relatively impermeable to the mobile ions in the thick-channel device,

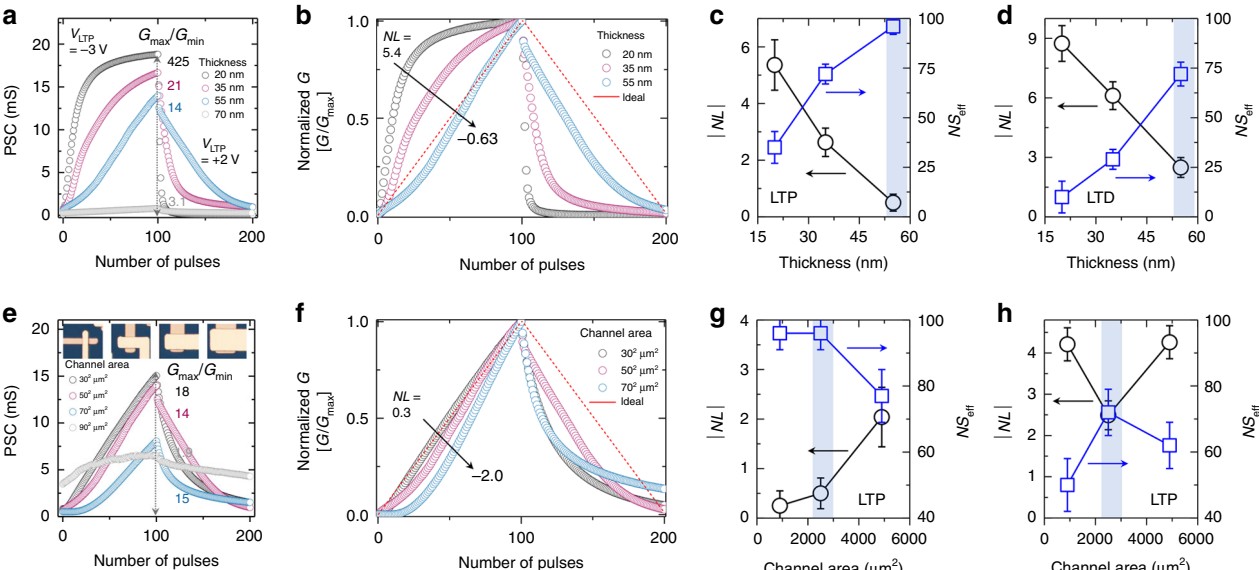

**Fig. 3 Optimization of vertical synaptic device geometry. a** LTP/D characteristics of thickness-controlled vertical crossbar synaptic devices under application of 100/100 potentiation/depression pulses ($V_{WC}$ pulses with amplitudes of −3 V/+2 V). Devices had P3HT channels with thicknesses of 20, 35, 50, and 70 nm. **b**, Normalized LTP/D curves of thickness-controlled synaptic devices. **c** Plots of |NL| and $NS_{eff}$ as functions of thickness of P3HT channel in LTP region and (d) LTD region. **e** LTP/D characteristics of synaptic devices with various areas of line cross-point. Channel area ($A_{ch}$) was controlled to $30 \times 30$, $50 \times 50$, $70 \times 70$, and $90 \times 90$ μm² as shown in the inset. **f** Normalized LTP/D curves of area-controlled synaptic devices. Plots of |NL | and $NS_{eff}$ as functions of area of line cross-point in **g** LTP and **h** LTD region. The average |NL|and $NS_{eff}$ are obtained from five devices prepared independently, and the error bars represents the standard deviation of the data.

which degraded the controllability of channel conductance under the same number of $V_{WC}$ pulses. Note that a similar behavior of the NL value was seen in the LTD region. The synaptic device with the 20-nm-thick channel exhibited a sharp decrease in the NL value as high as $-8.7$, and the 55-nm-thick P3HT exhibited a relatively linear decreasing characteristic ($NL_{LTD} = -2.3$). Figure 3c, d show the absolute NL and $NS_{eff}$ values plotted as functions of the P3HT thickness for the LTP and LTD characteristics, respectively. Here, states having $\Delta G$ above the noise range (0.5% of $G_{max} - G_{min}$) were defined as the effective states. The $|NL|$ value (denoted by black circles) was lowest (0.6/2.3 for LTP/D regions) for the device with the 55-nm-thick P3HT channel. This device also showed the highest $NS_{eff}$ of 96/72 for the LTP/D regions.

Next, the effect of the channel area on the LTP/D characteristics was investigated by varying the width of metal lines (Fig. 3e, f). The channel areas were controlled to $30 \times 30$, $50 \times 50$, $70 \times 70$, and $90 \times 90$ $\mu m^2$, where the thickness of the P3HT channel (that is, channel length) was fixed at 55 nm. Among these devices, the one with the largest channel area ($90 \times 90$ $\mu m^2$) showed the highest conductance value in the absence of any external voltage stimulus ($V_{WC} = 0$ V), which indicates that the device had the lowest channel resistance (Supplementary Fig. 7). However, this device with the $90 \times 90$ $\mu m^2$ channel showed a poor $G_{max}/G_{min}$ of 1.9, which was attributed to the obstruction of ion penetration by the large metal coverage (Supplementary Fig. 8). The $G_{max}/G_{min}$ value increased from 1.9 to 18 as the channel area decreased from $90 \times 90$ $\mu m^2$ to $30 \times 30$ $\mu m^2$, because of the enhanced channel controllability. The highest $G_{max}$ and $G_{max}/G_{min}$ values were obtained for the device with the $30 \times 30$ $\mu m^2$ channel. The absolute value of NL and the $NS_{eff}$ value for the LTP/D regions were plotted as a function of the channel area, as shown in Fig. 3g, h. The device with a channel area of $50 \times 50$ $\mu m^2$ exhibited desirable synaptic properties in both the LTP and LTD regions (i.e., low $|NL|$ values and the highest $NS_{eff}$). The device with the channel area of $30 \times 30$ $\mu m^2$ also showed a low $|NL|$ of 0.35 and high $NS_{eff}$ of 96. However, this device had a high $|NL|$ value of 4.2 and a low $NS_{eff}$ of 50 in the LTD region. Overall, the LTP/D characteristics of the ion-gel-gated vertical synapse were strongly affected by the ion penetration into the organic channel. From these results, the device with the channel area of $50 \times 50$ $\mu m^2$ and channel thickness of 55 nm was confirmed to exhibit desirable LTP/D characteristics such as large $G_{max}/G_{min}$, low $|NL|$, and sufficient $NS_{eff}$.

**Operational stability of vertical synaptic device.** In addition to the high LTP/D performance of a synaptic device, its operation must also be stable to enable its practical application to an ANN[44,45]. To investigate the repeatability and stability of the LTP/D characteristics in each cycle, various pulse sets with different numbers of pulses were applied to the WC terminal. The potentiation and depression pulses were set to $-3$ V and $+2$ V, respectively (see the LTD optimization procedure in Supplementary Fig. 9 and Supplementary Note 5). Figure 4a shows the PSC response of the synaptic device over five cycles under the application of different numbers of pulses, ranging from 5 to 100 (a total of 2000 pulses). The LTP/D characteristic curves extracted under the application of various pulse sets were highly stable and repeatable over the five cycles. To investigate the synaptic characteristics more quantitatively, we calculated the $|NL|$ and $NS_{eff}$ values. As shown in Fig. 4b, our synaptic device showed a linear weight-update behavior of $|NL| < 2$ for all pulse sets. We then extended the number of measurement cycles to 50 (a total of 10000 update pulses) and

investigated the PSC response (Fig. 4c and Supplementary Fig. 10). During the test, our device showed reliable LTP/D behavior without any sign of degradation (Fig. 4d). Furthermore, key parameters such as NL, $NS_{eff}$, and $G_{max}/G_{min}$ remained constant during the lengthy cycle test (Fig. 4e). The cycle-to-cycle variations of $|NL|$, $NS_{eff}$, and $G_{max}/G_{min}$ were calculated to be 1.4%, 6.4%, and 1.1%, respectively. Additionally, we investigated the LTP/D characteristics by elongating the cycle test to 200 k pulses for observing the degradation of the device performance (Supplementary Fig. 11 and Supplementary Note 6). Then, reliability of the PSC was also investigated under irregular-pulse conditions. Figure 4f shows a plot of the real-time change in the PSC of the device under the application of two different $V_{WC}$ pulse sets. The regular set consisted of successive potentiation (P) and depression (D) pulses (PPPDDD), whereas the random set consisted of randomly arranged P and D pulses (PPDPDD). For a detailed evaluation of the PSC change, the PSC plots in the first and last cycles of the regular set (black line) and random set (red line) were overlapped, as shown in the middle and right panels of Fig. 4g. The variation in the PSC state between the regular- and random-pulse conditions was lower than 1% at the base state for every pulse cycle. This value is quite comparable with other research results reported thus far (see the comparison in Supplementary Table 1 and Supplementary Note 7). Overall, the optimized vertical synaptic device showed stable weight-update behavior under various $V_{WC}$ conditions. Additionally, we investigated the writing energy of the device for single potentiation/depression pulse by measuring the current between WC terminal and postsynaptic terminal (see details in Supplementary Fig. 12 and Supplementary Note 8). The device exhibited energy consumption of 11.9/1.6 nJ for the potentiation/depression pulse, which was further reduced to 0.25/0.17 nJ under a pulse width of 700 ns.

**Logic application of vertical synaptic device and training/recognition processes.** Finally, to confirm the applicability of the developed device to HW-NNs, we prepared a small-scale synapse array with a size of $2 \times 3$ and trained Boolean logic operations such as binary AND and OR onto the neural network based on the synapse array. Various logic gate operations can be implemented on this neural network platform via the training of the gate functions, as in the case of reconfigurable circuits[1,43,46]. Figure 5a shows circuit diagrams of AND and OR gates composed with our synaptic devices. $V_1$ and $V_2$ denote two logic inputs, and $V_b$ denotes a bias voltage. The $G$ means a conductance value of the synaptic device. The $V_{WC1}$, $V_{WC2}$, and $V_{WCb}$ denote weight-control voltages. $I_{AND}$ and $I_{OR}$ are the output currents corresponding to the AND and OR logic operations, respectively. This circuit matches a single-layer neural network consisting of three input neurons, two output neurons, and six synapses connecting them, as shown in Fig. 5b. The learning process of this neural network is as follows. When the input voltages are applied to the input neurons, the output current vector ($\boldsymbol{I}$) is calculated as the inner product between the conductance matrix of the synaptic array ($\boldsymbol{G}$) and the input voltage vector ($\boldsymbol{V}$), i.e., $\boldsymbol{I} = \boldsymbol{GV}$. We then compare the obtained output currents with a threshold value (here, $I_{TH} = -5$ nA), consequently distinguishing whether the output is "0" or "1". If the output current is less than $-5$ nA, its state is considered as "0"; else, its state is considered as "1". As the final step of the learning, all conductance values in the synaptic array are updated by $V_{WC}$ toward reducing difference between the output value and the corresponding value of the truth table in Fig. 5c. Through this learning process, we updated all conductance values of the synaptic array in real-time and then

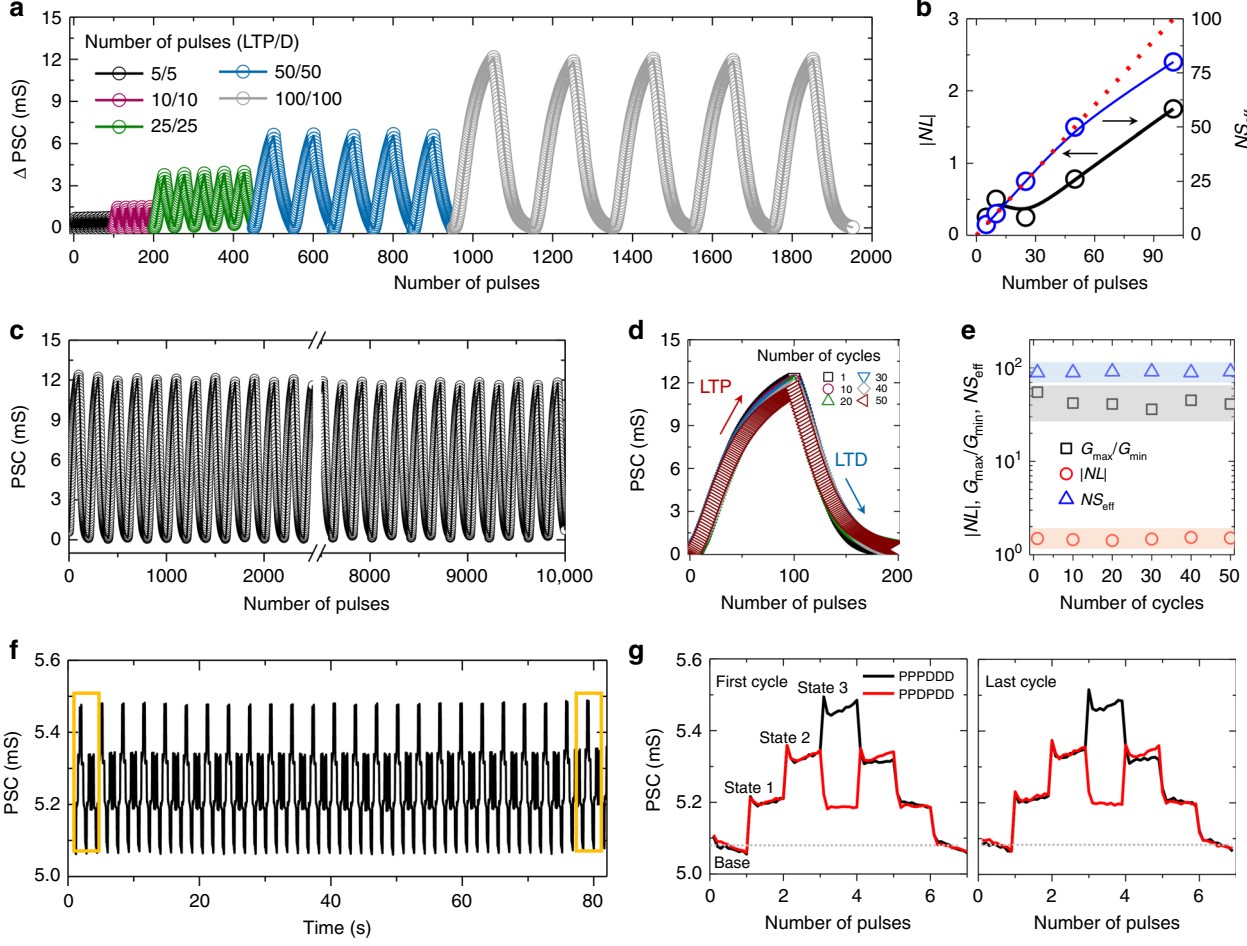

**Fig. 4 Operational stability of vertical synaptic device. a** LTP/D characteristics of vertical crossbar synapse array under application of various potentiation/depression pulse sets. **b** Plots of |NL| and $NS_{eff}$ as functions of pulse number for single cycle. **c** LTP/D characteristics of vertical synaptic device over 50 cycles. Number of potentiation/depression pulses for single cycle was set to 100/100. **d** Cycle-to-cycle variations of LTP/D curve for 50 cycles. **e** Plots of $G_{max}/G_{min}$, |NL|, and $NS_{eff}$ for 50 LTP/D cycles. **f** State stability under application of random combinations of potentiation/depression pulses with amplitudes of −3 V/+2 V. **g** Overlapping of PSC plots in the first and last cycles of regular (PPPDDD) and irregular (PPDPDD) pulse sets.

investigated output currents when the two logic inputs were "00", "10", "01", or "11", respectively (Fig. 5d). Two voltage values of 0.02 V and 0.2 V were used for binary logic input states of "0" and "1", respectively. Before training (black line), the initial output currents were close to the $I_{TH}$ so that were difficult to determine the output state. Particularly, when the inputs were "10" or "01", there were little differences between the output current values and the threshold value in both operations of AND and OR. However, as the learning proceeded and after completing the learning, the differences between those values became more explicit. This result successfully demonstrated that our synapse array can be functionalized as the AND and OR logic gates via the learning process.

To further investigate the feasibility of the synapse array toward HW-NNs, we theoretically constructed a large-scale NN with a size of 400 × 200 × 10 with the measured LTP/D characteristics of 10 synaptic devices (Fig. 5e)[44]. We then performed the training/recognition tasks for Modified National Institute of Standards and Technology (MNIST) digit patterns and plotted the recognition rate of each device at every 40,000 training number (1 epoch) in Fig. 5f. The corresponding parameters of each synaptic device used in the simulation are listed in Supplementary Table 2. Among the 10 devices, the maximum accuracy was recorded to be as high as 92.5% when the NL value for LTP/D and the dynamic range were −1.25/−5.72 and 10.72, respectively. The minimum

accuracy was 85.7% when the NL value for LTP/D and the dynamic range were −0.42/−6.77 and 49.33, respectively. Further improvement in the accuracy was achieved later by pulse engineering (Supplementary Fig. 13 and Supplementary Note 9). We then investigated the device-to-device variation in the recognition rate for the 10 synaptic devices (Fig. 5g). The standard deviation of maximum recognition rates was as low as 2.5%, and the standard deviation after the 25 epoch learning was 4.2%. Through this theoretical learning and recognition task, we confirmed the applicability of the proposed synapse array for more complex HW-NNs.

## Discussion

In this study, we successfully implemented a crossbar synapse array based on a vertical organic transistor with an ion-gel WC layer. This three-terminal synapse array was achieved by adopting the vertical gate-all-around field effect transistor (GAA-FET) concept and securing acceptable gate controllability with the assistance of an ion-gel dielectric. Mobile ions in the ion gel penetrated the free volume in the P3HT vertical organic channel located at every cross-point between the top and bottom electrode lines, which resulted in a nonvolatile change in the channel conductance. By virtue of ion movement, the proposed device exhibited diverse synaptic characteristics, such as STP, EPSC/IPSC, and LTP/D. In particular,

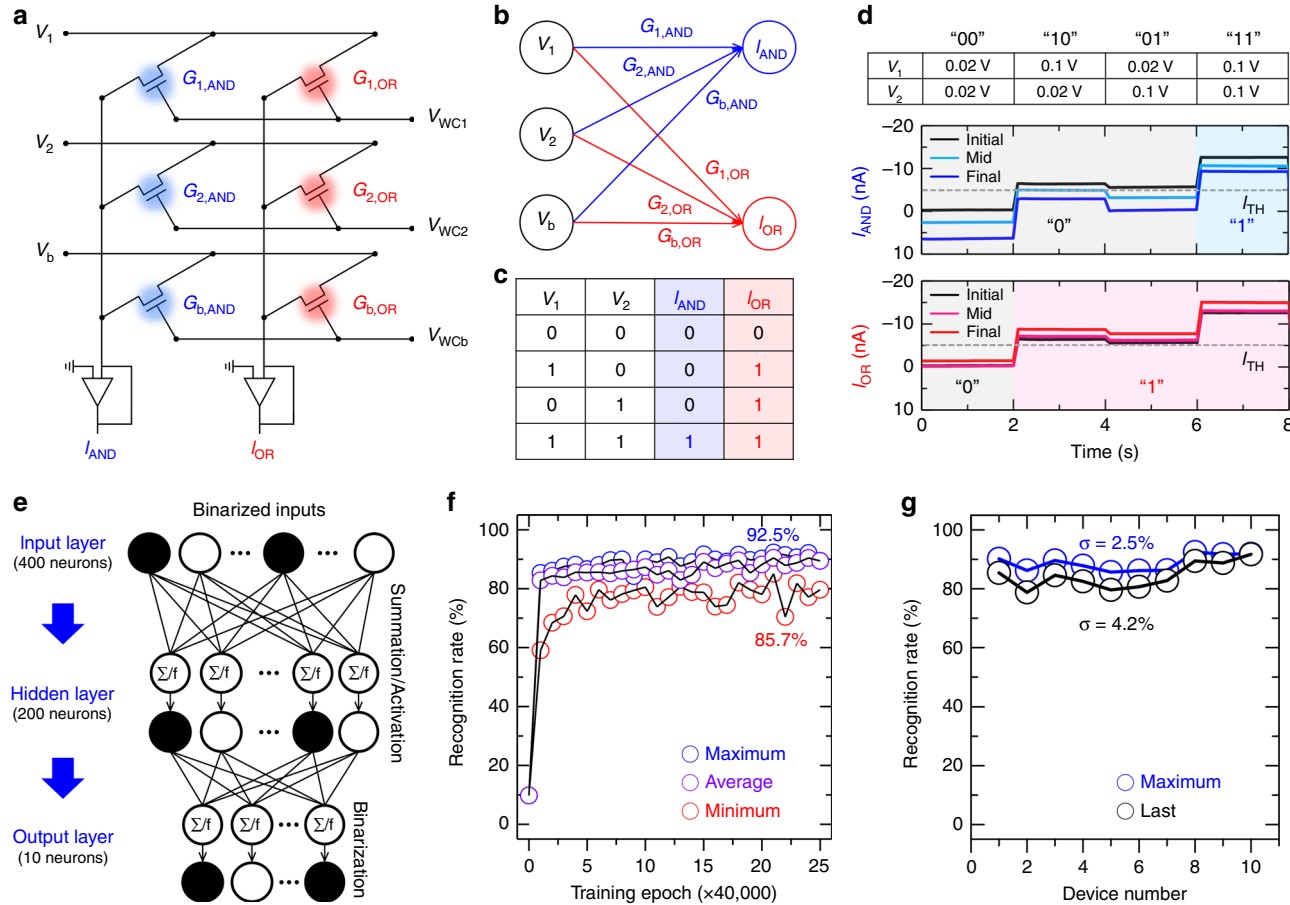

**Fig. 5 Logic application of vertical synaptic device and training/recognition processes. a** Schematic illustration of AND and OR logic gates implemented using proposed synaptic array with size of 2 × 3. **b** Simplified diagram of NN for AND and OR logic gates. **c** Truth table of AND and OR logic gates. **d** Real-time training and classification of AND and OR gates using implemented synaptic array. **e** Schematic illustration of two-layer perceptron-based ANN with size of 400 × 200 × 10. **f** Recognition rate as a function of number of training epochs for 10 synaptic devices; the maximum, average, and minimum recognition rates are indicated in blue, purple, and red, respectively. **g** Maximum (blue) and final (black) recognition rates of 10 synaptic devices.

optimization of the channel length and area of the line cross-point yielded excellent LTP/D characteristics, such as a large dynamic range (>10), low nonlinearity (<1), sufficient effective number of conductance states (>64), and low cycle variation (<1%). Furthermore, we demonstrated the feasibility of using the proposed vertical organic synapse array for implementing a complex NN through real-time training and classification tasks in a simple 2 × 3 NN. A very high recognition rate of 92.5% for MNIST digit patterns was achieved in a simulated two-layer ANN with a size of 400 × 200 × 10. To implement a hardware ANN with the vertical organic synapses as a follow-up research, the excellent endurance of the synapses is critically required. In this regard, identifying and understanding the failure mechanism for weight update will help in assessing and improving the endurance. Besides, the researches optimizing encapsulation layers, ion-gel dielectrics, and organic semiconductors in the synapses need to be done for the excellent endurance. Notably, this GAA-FET concept has already been considered for 3-nm technology nodes (for a lateral type) and next technology node (for a vertical type) by many global semiconductor companies. Thus, this research is meaningful as a proof-of-concept of a cross-point FET-type synapse array that can be used to implement NNs based on Si CMOS technology. We expect the proposed vertical crossbar synapse array to play a pioneering role in the development of high-performance and high-density NNs in the future.

## Methods

**Materials**. Processing solvents such as acetone, chloroform, and 2-propanol were purchased from Sigma-Aldrich. Regioregular P3HT ($M_n$: 54,000–75,000, lot number: MKCK1947), PVdF-HFP ($M_n$: 110,000), and 1-ethyl-3-methylimidazolium bis(trifluoromethylsulfonyl)imide (EMIM:TFSI) ionic liquid were also purchased from Sigma-Aldrich. The azide crosslinker bis(6-((4-azido-2,3,5,6-tetrafluorobenzoyl)oxy) hexyl)decanedioate was synthesized as previously reported[42]. P3HT solution was prepared by dissolving P3HT in concentrations of 3, 5, 7, 9, 11, and 13 mg mL$^{-1}$ in chloroform for obtaining P3HT organic channels of various thicknesses. After being stirred on a hot plate for 5 h at 50 °C, the P3HT solution was blended with the azide crosslinker (5 mg mL$^{-1}$ in chloroform) in a 4:1 ratio. Ion-gel solution was prepared by mixing the EMIM:TFSI ionic liquid, PVdF-HFP, and acetone solvent in a 4:1:7 ratio. The ion-gel solution was stirred at 50 °C for 5 h before use.

**Fabrication of vertical synapse**. The bottom metal lines (Cr/Au with 1 nm/17 nm thickness) were thermally deposited on a cleaned SiO$_2$/Si$^{++}$ (thermally grown 100 nm SiO$_2$) wafer. The photopatternable P3HT solution was spin-coated on top of the metal lines at 1,500 rpm for 30 s. The crosslinking reaction of the channel area was performed under selective UV irradiation (254 nm and 1000 W cm$^{-2}$) through a metal shadow mask for 30 s. Then, unexposed P3HT solution was removed with chloroform to define the crosslinked channel area, and the sample was dried for 12 h in a glovebox. After deposition of the top metal lines (30-nm-thick Au) via thermal evaporation, the ion-gel layer was spin-coated at 1000 rpm and dried at 70 °C for 5 min. Finally, 50-nm-thick Au as a WC line was thermally deposited.

**Device characterization**. The thickness of the crosslinked P3HT film was analyzed using tapping-mode atomic force microscopy (Asylum Cypher S system). The electrical properties of the vertical synaptic device and logic gates were measured using a Keithley 4200 electrometer.

***NL calculation***. The NL value of the LTP/D curve was calculated using the following equations:

$$G_{LTP} = B \cdot (1 - \exp(-P/A_P)) + G_{min}, \quad (1)$$

$$G_{LTD} = -B \cdot (1 - \exp((P - P_{max})/A_D)) + G_{max}, \quad (2)$$

$$B = (G_{max} - G_{min})/\left(1 - \exp\left(-P_{max}/A_{P,D}\right)\right) \quad (3)$$

where $G_{LTP}$ and $G_{LTD}$ are the conductance values of the LTP and LTD regions, respectively; $P$ is the number of applied pulses; $A$ is a parameter representing NL; and $B$ is a fitting constant used to normalize the conductance range. The $A$ value was extracted from the experimental data using the MATLAB code provided as an open source[44], and the corresponding NL values were derived from tables provided by the same source.

**Simulation of two-layer neural network**. The simulation was conducted on the basis of the platform "MLP + NeuroSim ver. 1.0". A multilayer (two-layer) perceptron-based ANN with a size of $400 \times 200 \times 10$ was theoretically constructed using non-ideal factors, including the dynamic range and NL. Then, $20 \times 20$ MNIST digit patterns were binarized to black-and-white patterns, and a logistic function was used as the activation function. The optimized learning rates for the first and second synaptic weight matrices were 0.2 and 0.025, respectively. After being trained with 1 million patterns, the two-layer ANN was used to perform a classification task for 10,000 separate testing images. The recognition rate was calculated for every 40,000 images during the training process.

## Data availability

All data generated or analyzed during this study are included in this published article (and its Supplementary Information files).

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

## Acknowledgements

We acknowledge the grants from Basic Science Research Program and Nano Material Technology Development Program through the National Research Foundation of Korea

(NRF) funded by the Ministry of Science, ICT & Future Planning (2020R1A2C2007819, 2016M3A7B4910426, and 2020R1A4A2002806) and the Creative Materials Discovery Program (NRF-2019M3D1A1078299) through the NRF of Korea funded by the Ministry of Science and ICT, Korea.

## Author contributions

Y.C. and S.O. contributed equally to this work as the first author. Y.C. conducted the fabrication and analysis of the device. S.O. conducted the equipment setup for measuring synaptic characteristics as well as construction and evaluation of the ANN. C.Q. gave a general advice on measuring and analysis of synaptic properties. J.H.P. and J.H.C. initiated the research, designed all the experiments. All authors discussed the results and contributed to the paper.

## Competing interests

The authors declare no competing interests.
