## [Peer Review File · Nature Communications]

Reviewers' Comments:

Reviewer #1:

Remarks to the Author:

The authors explored novel vertical synaptic devices with remote synaptic weight update via ion gel. These devices could form a crossbar array for further integration. The authors systematically investigated the influence of channel length and area on channel conductance so as to achieve the optimal LTP/D characteristics. Moreover, they showed the applicability of their synaptic devices in a neural network. The current work may contribute to the development of advanced neural networks given the combined advantages of both two- and three-terminal synaptic devices. However, I would like to ask the authors to address the following issues before the publication of this work.

1. In Figure 2b and e, the authors only discussed the nonlinearity (NL) and effective number of states (N_{Seff}) of the LTP regions. How about the LTD regions? Why are the LTD regions in Figure 2b are nearly the same? Please add discussion in the manuscript.
2. The authors chose A to evaluate the NL. However, other researchers used the asymmetric ratio (AR) [e. g., Yang, C. S., et al., *Adv. Funct. Mater.* 28, 1804170, (2018); Yin, L. et al., *Nano Energy* 63, 103859 (2019)]. Could the authors clearly justify their current method to evaluate the NL in the manuscript?
3. The cycle-to-cycle variations of the LTP/D characteristics were investigated. Please compare the current variations with those reported before and comment on whether they are suitable for a hardware neural network in the manuscript.
4. The authors demonstrated Boolean logic operations by using a small-scale array. Is the realization of logic operations only due to the intrinsic electrical properties of a transistor? Do the synaptic functionalities of the devices play any role in logic operation? Please provide comments in the manuscript.
5. Given the large-scale deployment of neuromorphic computing in the future, it may be a good choice to take advantage of the mature silicon processing. The authors may state the implication of the current work for neuromorphic computing based on silicon where appropriate.

Reviewer #2:

Remarks to the Author:

The authors present a novel study ranging from vertical organic synaptic device to artificial neural network. The proposed vertical ion-gel-gated synapse could achieve a remote weight update via gate driven ion penetration in P3HT channel, allowing a 3D crossbar array based on three-terminal synaptic cells. The authors provide a generalized approach to significantly improve packing density of three-terminal synaptic devices. Additionally, the ion-gel-gated synapse show intriguing device properties including low nonlinearity (<1), large dynamic range (>10), high endurance and reproducibility. This study is innovative in device architecture, and clear in its theme. Despite I am positively impressed by structural innovation and experiment result, several critical points must be addressed to enhance the manuscript for publishing in high-impact Nature Communications.

1. The authors claim that the conductance control mechanism is based on gate driven ion transfer between ion gel and P3HT, while, mechanism research and experimental verification are missing.
2. As indicated in Figure 1d, more significant conductance changes could be induced by negative V_{wc} , Why? The authors should comment. LTP/LTD characteristics were achieved by 100 consecutive potentiation pulses ($V_{\text{wc}} = -3 \text{ V}$) followed by 100 consecutive depression pulses ($V_{\text{wc}} = +2 \text{ V}$) in Figure 1f; while, in Figure 2 and Figure 3, the authors used V_{wc} pulses with amplitudes

of ± 3 V. Please comment.

3. Symmetrically programmable conductance states are desirable for facilitating 'blind' synaptic weight update during learning. Symmetry of conductance tuning remains to be improved in this study. The revelation of the physical mechanism (point 1) may be helpful for further device optimization.

4. The font (red) in Figure 1c is too small to read. Please modify.

5. The reviewer recommends the authors to include "Adv. Mater. 2018, 30, 1802883; Adv. Funct. Mater. 2019, 29, 1902374" as references for "three-terminal artificial synapses implemented with various inorganic and organic materials....." (Page 2, line 51).

6. What does WC stand for? The authors should give the full name of the abbreviation.

7. Fixing device variability is a prerequisite for mass production. What about the device-to-device variability.

Reviewer #3:

Remarks to the Author:

The paper presents a type of ionic transistor, amenable to 3d integration, and demonstrates the use in training neural networks. The device structure's compatibility with efficient 3d integration, combined with enhanced analog capabilities of an ionic device are the main novel features claimed in the work. This device structure is novel, and the work is original, and this is a topic of significant recent interest. However, the characteristics are not competitive with that found in top state of the art published results. I do not find the results impactful enough to recommend publication in Nature Communications. A future version of this work might be suitable for publication, if a number of issues are addressed:

- 92.5% accuracy on MNIST training does not represent state of the art. In state of the art devices, the full precision matches the ionic device training accuracy which is about 98% (see ref 1). Also, the conductance versus pulse width curves show nice linearity for many of the increasing conductance cases/LTP, but all curves show a very sharp nonlinear drop for decreasing conductance/LTD. The plots of NL in Fig. 2c and 2f appear only to take into account the increasing cases/LTP. In our experience, highly nonlinear LTD coupled with asymmetric LTP versus LTD precludes reasonable accuracy for training even simple datasets like MNIST. Was this LTD nonlinearity included in the neurosim modeling and still a 92.5% accuracy in Fig. 4? If this accuracy was obtained even with nonlinear decrease, then the authors should discuss if a scheme was used to mitigate asymmetric nonlinearity (because it is surprising that neurosim would give even this moderately high MNIST accuracy with that level of LTD NL/asymmetry without a mitigation scheme).

- The STP investigation appears to show a state decay on the order of seconds. Although this is useful for STP, it is a problem for neural network training. If the weights are quickly decaying significantly during the time of a few training epochs, this will degrade the accuracy on benchmark training, such as MNIST. Please discuss if and how this drift was accounted for in the theoretical study of the MNIST training described on pg 8 and Fig. 4.

- Writing the device requires a pulse width of 50ms. This is far too slow to be useful for the VMM-type applications of interest (for training a deep network), such as those discussed in references 1, 9, and 18. Related work has been able to show 200ns programming (i.e. reference 1). Is there a path, perhaps through scaling for the proposed system to achieve sub-microsecond write speeds?

- Is 10k pulses the maximum number of cycles before degradation occurs? Is there a reason not to measure out to a greater number of cycles? It seems more than 10k pulses will be needed even to train MNIST, and if there is degraded analog performance later in the training process this should be accounted for in the model.

- Write currents and energies need to be discussed. As discussed in many of this article's references, low energy DNN training is a key advantage for analog, and a key reason we are interested in these new devices. With the long write time, if the device requires even 1uA, the write energy will be on the order of $50\text{ms} \times 1\mu\text{A} \times 3\text{V} = 150\text{nJ}$. This is several orders of magnitude higher than the energy of $<10\text{pJ}$ published in recent works. If the write energy is in fact very high,

a path to scaling and significantly reducing it should be discussed as well.

- The 3D compatible process is one of the potential advantages of this device. However, as with other similar recently demonstrated ionic analog devices, it would still need to be integrated with CMOS to control the arrays. The authors did not comment or consider CMOS compatibility. It seems that some of the materials, such as the P3HT would probably not be able to withstand the CMOS back end of line temperatures (typically about 400C). Would this be a likely showstopper for this device?

Point-by-Point Response to the Reviewers

Reviewer #1 (Remarks to the Author):

The authors explored novel vertical synaptic devices with remote synaptic weight update via ion gel. These devices could form a crossbar array for further integration. The authors systematically investigated the influence of channel length and area on channel conductance so as to achieve the optimal LTP/D characteristics. Moreover, they showed the applicability of their synaptic devices in a neural network. The current work may contribute to the development of advanced neural networks given the combined advantages of both two- and three-terminal synaptic devices. However, I would like to ask the authors to address the following issues before the publication of this work.

ANS: Thank you for reviewing our paper. We appreciate your insightful comments on our research. We have revised the manuscript according to your suggestions and believe that these revisions have improved the paper.

Please find below our responses (in blue) to each of your specific comments (in black). Revisions to the original article are indicated in red.

(1) In Figure 2b and e, the authors only discussed the nonlinearity (NL) and effective number of states (NS_{eff}) of the LTP regions. How about the LTD regions? Why are the LTD regions in Figure 2b are nearly the same? Please add discussion in the manuscript.

ANS: As the reviewer mentioned about the previous **Figure 2b**, G/G_{max} rapidly decreased when the first 10 depression pulses were applied, showing very similar LTD characteristics (NL and NS_{eff}) regardless of the vertical channel length (thickness). This is because the mobile ions penetrating the P3HT channel were abruptly moved to the ion-gel region by the large depression voltage pulses with amplitude of 3 V.

For the following measurements, we reduced the amplitude of the depression pulses to 2 V and investigated the LTP/D characteristics again. As a result, as shown in the new **Figure 2b** (see below), the NL and NS_{eff} values for LTP varied from 5.4 to -0.63 and from 10 to 72 as the channel length increased from 20 to 55 nm in the LTP region. A similar trend was observed in the LTD region. In addition, the previous version of **Figure 2e** was revised with the LTP/D curves measured under $V_{\text{LTP/D}} = -3 \text{ V}/+2 \text{ V}$.

We prepared a new **Figure 2** and provided the relevant information in the manuscript as shown below.

Fig. 2 Optimization of vertical synaptic device geometry a, LTP/D characteristics of thickness-controlled vertical crossbar synaptic devices under application of 100/100 potentiation/depression pulses (V_{WC} pulses with amplitudes of $-3 \text{ V}/+2 \text{ V}$). Devices had P3HT channels with thicknesses of 20, 35, 50, and 70 nm. **b**,

Normalized LTP/D curves of thickness-controlled synaptic devices. **c**, Plots of $|NL/|$ and NS_{eff} as functions of thickness of P3HT channel in LTP region and (d) LTD region. **e**, LTP/D characteristics of synaptic devices with various areas of line cross-point. Channel area (A_{ch}) was controlled to 30×30 , 50×50 , 70×70 , and $90 \times 90 \mu\text{m}^2$ as shown in the inset. **f**, Normalized LTP/D curves of area-controlled synaptic devices. Plots of $|NL/|$ and NS_{eff} as functions of area of line cross-point in **g**, LTP and **h**, LTD region. The average $|NL/|$ and NS_{eff} are obtained from 5 devices prepared independently, and the error bars represents the standard deviation of the data.

In the manuscript:

“**Fig. 2a and Supplementary Figure 3** show the LTP/D characteristics of vertical synaptic devices with P3HT channels of various thicknesses under the application of a set of V_{WC} pulses consisting of 100 potentiation pulses ($V_{\text{WC}} = -3 \text{ V}$) and 100 depression pulses ($V_{\text{WC}} = +2 \text{ V}$). Here, the pulse frequency and pulse width were fixed at 2 Hz and 50 ms, respectively (see **Supplementary Figures 4 and 5** for the additional information about various pulse frequencies and pulse widths). The thickness of the P3HT channel was controlled in the range of 20–95 nm by varying the concentration of the P3HT solution, whereas the channel area was fixed at $50 \times 50 \mu\text{m}^2$. The maximum conductance value (G_{max}) was the highest (18.9 mS) in the device with the thinnest channel (20 nm), and it decreased to 2.3 μS with an increase in the thickness of the P3HT channel to 95 nm. Because the channel thickness is considered as the channel length in a vertically stacked device, in this study, the channel length of the synaptic device increased with increasing thickness of the P3HT channel; this resulted in a decrease in the overall channel conductance. The dynamic range ($G_{\text{max}}/G_{\text{min}}$) values of the synaptic devices having 20-, 35-, and 55-nm-thick P3HT channels were higher than 10, which is the minimum value required for a successful pattern recognition task⁴⁵. To extract the values of NL and NS_{eff} , which represent the precision of the weight-update behavior, we first normalized the LTP/D characteristic curves of the devices with different channel lengths by dividing each conductance value by the maximum value (G/G_{max}), as shown in **Fig. 2b**. Then, we calculated the NL value by fitting the measured curve to the normalized one (see detailed equations in the METHODS section and **Supplementary Figure 6**). The synaptic device with the thinnest channel (20 nm) showed a positive NL value (+5.4). In this device, the ions were able to penetrate the entire channel upon the application of V_{WC} pulses, which enhanced the overall channel controllability. In contrast, the NL values of the devices with the thicker channels decreased in the negative direction and reached -0.6 for the device with the 55-nm-thick P3HT channel. This is because the channel region far away from the ion gel was relatively impermeable to the mobile ions in the thick-channel device, which degraded the controllability of channel conductance under the same number of V_{WC} pulses. Note that a similar behavior of the NL value was seen in the LTD region. The synaptic device with the 20-nm-thick channel exhibited a sharp decrease in the NL value as high as -8.7 , and the 55-nm-thick P3HT exhibited a relatively linear decreasing characteristic ($NL_{\text{LTD}} = -2.3$). **Fig. 2c and 2d** show the absolute NL and NS_{eff} values plotted as functions of the P3HT thickness for the LTP and LTD characteristics, respectively. Here, states having ΔG above the noise range (0.5% of $G_{\text{max}} - G_{\text{min}}$) were defined as the effective states. The $|NL/|$ value (denoted by black circles) was lowest (0.6/2.3 for LTP/D regions) for the device with the 55-nm-thick P3HT channel. This device also showed the highest NS_{eff} of 96/72 for the LTP/D regions.

Next, the effect of the channel area on the LTP/D characteristics was investigated by varying the width of metal lines (**Fig. 2e and 2f**). The channel areas were controlled to 30×30 , 50×50 , 70×70 , and $90 \times 90 \mu\text{m}^2$, where the thickness of the P3HT channel (that is, channel length) was fixed at 55 nm. Among these devices, the one with the largest channel area ($90 \times 90 \mu\text{m}^2$) showed the highest conductance value in the absence of any external voltage stimulus ($V_{\text{WC}} = 0 \text{ V}$), which indicates that the device had the lowest channel resistance (**Supplementary Figure 7**). However, this device with the $90 \times 90 \mu\text{m}^2$ channel showed a poor $G_{\text{max}}/G_{\text{min}}$ of 1.9, which was attributed to the obstruction of ion penetration by the large metal coverage (**Supplementary Figure 8**). The $G_{\text{max}}/G_{\text{min}}$ value increased from 1.9 to 18 as the channel area decreased from $90 \times 90 \mu\text{m}^2$ to $30 \times 30 \mu\text{m}^2$, because of the enhanced channel controllability. The highest G_{max} and G_{min} values were obtained for the device with the $30 \times 30 \mu\text{m}^2$ channel. The absolute value of NL and the NS_{eff} value for the LTP/D regions were plotted as a function of the channel area, as shown in **Fig. 2g and 2h**. The device with a channel area of $50 \times 50 \mu\text{m}^2$ exhibited desirable synaptic properties in both the LTP and LTD regions (i.e., low $|NL/|$ values and the highest NS_{eff}). The device with the channel area of $30 \times 30 \mu\text{m}^2$ also showed a low $|NL/|$ of 0.35 and high NS_{eff} of 96. However, this device had a high $|NL/|$ value of 4.2 and a low NS_{eff} of 50 in the LTD region. Overall, the LTP/D characteristics of the ion-gel-gated vertical synapse

were strongly affected by the ion penetration into the organic channel. From these results, the device with the channel area of $50 \times 50 \mu\text{m}^2$ and channel thickness of 55 nm was confirmed to exhibit desirable LTP/D characteristics such as large $G_{\text{max}}/G_{\text{min}}$, low $|NL|$, and sufficient NS_{eff} .”

(2) The authors chose A to evaluate the NL. However, other researchers used the asymmetric ratio (AR) [e.g., Yang, C. S., et al., *Adv. Funct. Mater.* 28, 1804170, (2018); Yin, L. et al., *Nano Energy* 63, 103859 (2019)]. Could the authors clearly justify their current method to evaluate the NL in the manuscript?

ANS: As the reviewer mentioned, there are several methods to evaluate the nonlinearity (NL) of the LTP/LTD characteristic curve^[R1-4]. Among them, we chose a method to tune A_P and A_D for finding the G_{LTP}/G_{LTD} curves best matched to the measured LTP/LTD curves^[R3,4]. The G_{LTP}/G_{LTD} curve model with the number of pulses (P) is represented as the following equations:

$$G_{LTP} = B \cdot (1 - \exp(-P/A_P)) + G_{\min}, \quad (1)$$

$$G_{LTD} = -B \cdot (1 - \exp((P - P_{\max})/A_D)) + G_{\max}, \quad (2)$$

$$B = (G_{\max} - G_{\min}) / (1 - \exp(-P_{\max}/A_{P,D})) \quad (3)$$

G_{LTP} and G_{LTD} are the conductance values for LTP and LTD, respectively. G_{\max} , G_{\min} , and P_{\max} are measured data that represent the maximum conductance, minimum conductance, and maximum pulse number, respectively. B is a fitting constant to normalize the conductance range. A_P and A_D are parameters that determine the nonlinearities of the weight update in the LTP and LTD regions, which are directly related to the NL values^[R3].

The G_{LTP}/G_{LTD} curves with respect to NL ranging from 0 to 5 are shown in **Figure R1a**. By adjusting the A_P and A_D values, the G_{LTP}/G_{LTD} curves are fitted to the measured LTP/LTD curves, and accordingly the NL values are determined (**Figure R1b**).

Figure R1. Nonlinearity analysis of long-term potentiation (LTP) and long-term depression (LTP) characteristic curves. (a) G_{LTP}/G_{LTD} curves with respect to nonlinearity (NL) ranging from 0 to 5. (b) Measured/fitted curves in LTP (upper panel) and LTD (lower panel) regions, where $NL_{LTP/D} = -0.63/-2.27$.

[R1] Yang, C. S. et al. All-solid state synaptic transistor with ultralow conductance for neuromorphic computing. *Adv. Funct. Mater.* 28, 1804170 (2018).

[R2] Yin, L. et al. Synaptic silicon-nanocrystal phototransistors for neuromorphic computing. *Nano Energy* 63, 103859 (2019).

[R3] Chen, P., Peng, X. & Yu, S. Neurosim+: An integrated device-to-algorithm framework for benchmarking synaptic devices and array architectures. In: *2017 IEEE International Electron Devices Meeting (IEDM)* 6.1.1–6.1.4 (2017).

[R4] Qian, C. et al. Solar-stimulated optoelectronic synapse based on organic heterojunction with linearly potentiated synaptic weight for neuromorphic computing. *Nano Energy* 66, 104095 (2019).

To reflect the reviewer's comment, we added detailed information on the NL extraction to the Supporting Information as below.

In the Supporting Information:

“

Supplementary Figure 6 *NL* analysis of LTP and LTD characteristic curves **a**, G_{LTP}/G_{LTD} curves with respect to *NL* ranging from 0 to 5. **b**, Measured/fitted curves in LTP (upper panel) and LTD (lower panel) regions, where $NL_{LTP/D} = -0.63/-2.27$.

There are several methods to evaluate the *NL* of the LTP/LTD characteristic curve¹⁻⁴. Among them, we chose a method to tune A_P and A_D for finding the G_{LTP}/G_{LTD} curves best matched to the measured LTP/LTD curves^[3,4]. The G_{LTP}/G_{LTD} curve model with the number of pulses (P) is represented as the following equations:

$$G_{LTP} = B \cdot (1 - \exp(-P/A_P)) + G_{\min}, \quad (1)$$

$$G_{LTD} = -B \cdot (1 - \exp((P - P_{\max})/A_D)) + G_{\max}, \quad (2)$$

$$B = (G_{\max} - G_{\min}) / (1 - \exp(-P_{\max}/A_{P,D})) \quad (3)$$

where G_{LTP} and G_{LTD} are the conductance values for LTP and LTD, respectively. G_{\max} , G_{\min} , and P_{\max} are the measured data that represent the maximum conductance, minimum conductance, and maximum pulse number, respectively. B is a fitting constant to normalize the conductance range. A_P and A_D are parameters that determine the nonlinearities of the weight update in the LTP and LTD regions, which are directly related to the *NL* values³.

The G_{LTP}/G_{LTD} curves with respect to the *NL* ranging from 0 to 5 are displayed in **Supplementary Figure 6a**. By adjusting the A_P and A_D values, the G_{LTP}/G_{LTD} curves are fitted to the measured LTP/LTD curves, and accordingly the *NL* values are determined (**Supplementary Figure 6b**).

Supplementary References

1. Yang, C. S. et al. All-solid state synaptic transistor with ultralow conductance for neuromorphic computing. *Adv. Funct. Mater.* **28**, 1804170 (2018).
2. Yin, L. et al. Synaptic silicon-nanocrystal phototransistors for neuromorphic computing. *Nano Energy* **63**, 103859 (2019).
3. Chen, P., Peng, X. & Yu, S. Neurosim+: An integrated device-to-algorithm framework for benchmarking synaptic devices and array architectures. In: *2017 IEEE International Electron Devices Meeting (IEDM)* 6.1.1-6.1.4 (2017).
4. Qian, C. et al. Solar-stimulated optoelectronic synapse based on organic heterojunction with linearly potentiated synaptic weight for neuromorphic computing. *Nano Energy* **66**, 104095 (2019).

(3) The cycle-to-cycle variations of the LTP/D characteristics were investigated. Please compare the current

variations with those reported before and comment on whether they are suitable for a hardware neural network in the manuscript.

ANS: We summarized recent papers that studied how the cycle-to-cycle variation of the LTP/LTD characteristics affects the recognition rate for MNIST digit patterns (see **Table R1**). When the variation was under 3%^[R6,8], the corresponding recognition rates were higher than 90%. On the contrary, for a high variation above 3%^[R5,7], the recognition rate was seriously degraded below 90%.

Compared to these devices, our vertical organic synapse showed a very low cycle-to-cycle variation of less than 1% and thereby a high recognition rate of 92.5%. From the perspective of cycle-to-cycle variation, we can say that the proposed device is acceptable for implementing a hardware neural network.

Table R1. Recent studies on the cycle-to-cycle variation and the corresponding recognition rate for MNIST digit images.

	This work	AlO _x /HfO ₂ RRAM ^[S5]	SiGe epiRAM ^[S6]	a -MoO ₃ TFT ^[S7]	IGZO FeFET ^[S8]
Cycle-to-cycle variation	< 1%	5%	< 1%	6.5%/9.3% (LTP/LTD)	2.36%
Recognition rates (real/ideal devices)	92.5%/96.2%	41%/96%	95.1%/97%	87.3%/96.7%	91.1%/94.1%

[R5] Woo, J. et al. Improved synaptic behavior under identical pulses using AlO_x/HfO₂ bilayer RRAM array for neuromorphic systems. *IEEE Electron Device Lett.* **37**, 994–997 (2016).

[R6] Choi, S. et al. SiGe epitaxial memory for neuromorphic computing with reproducible high performance based on engineered dislocations. *Nat. Mater.* **17**, 335–340 (2018).

[R7] Yang, C. S. et al. All-solid state synaptic transistor with ultralow conductance for neuromorphic computing. *Adv. Funct. Mater.* **28**, 1804170 (2018).

[R8] Kim, M. K. & Lee, J. S. Ferroelectric Analog Synaptic Transistors. *Nano Letters* **19**, 2044–2050 (2019).

We added the relevant information to the manuscript and Supporting Information as below.

In the manuscript:

“The variation in the PSC state between the regular- and random-pulse conditions was lower than 1% at the base state for every pulse cycle. **This value is quite comparable with other research results reported thus far (see the comparison in Supplementary Table 1).** Overall, the optimized vertical synaptic device showed stable weight-update behavior under various V_{WC} conditions.”

In the Supporting Information:

“**Supplementary Tables**

Supplementary Table 1. Recent studies on the cycle-to-cycle variation and the corresponding recognition rate for MNIST digit images.

	This work	AlO _x /HfO ₂ RRAM ⁵	SiGe epiRAM ⁶	a -MoO ₃ TFT ⁷	IGZO FeFET ⁸
Cycle-to-cycle variation	< 1%	5%	< 1%	6.5%/9.3% (LTP/LTD)	2.36%
Recognition rates (real/ideal devices)	92.5%/96.2%	41%/96%	95.1%/97%	87.3%/96.7%	91.1%/94.1%

We summarized recent papers studying how the cycle-to-cycle variation of the LTP/LTD characteristics affects the recognition rate for MNIST digit patterns. When the variation was under 3%^{6,8}, the corresponding recognition rates were higher than 90%. On the contrary, for a high variation above 3%^{5,7}, the recognition rate was seriously degraded below 90%.

Compared to these devices, our vertical organic synapse showed a very low cycle-to-cycle variation of less than 1% and thereby a high recognition rate of 92.5%. From the perspective of cycle-to-cycle variation, it is evident that the proposed device is acceptable for implementing a hardware neural network.

5. Woo, J. et al. Improved synaptic behavior under identical pulses using $\text{AlO}_x/\text{HfO}_2$ bilayer RRAM array for neuromorphic systems. *IEEE Electron Device Lett.* **37**, 994–997 (2016).

6. Choi, S. et al. SiGe epitaxial memory for neuromorphic computing with reproducible high performance based on engineered dislocations. *Nat. Mater.* **17**, 335–340 (2018).

7. Yang, C. S. et al. All-solid state synaptic transistor with ultralow conductance for neuromorphic computing. *Adv. Funct. Mater.* **28**, 1804170 (2018).

8. Kim, M. K. & Lee, J. S. Ferroelectric Analog Synaptic Transistors. *Nano Lett.* **19**, 2044–2050 (2019).”

(4) The authors demonstrated Boolean logic operations by using a small-scale array. Is the realization of logic operations only due to the intrinsic electrical properties of a transistor? Do the synaptic functionalities of the devices play any role in logic operation? Please provide comments in the manuscript.

ANS: Before the training, the initial output currents (black line) had values close to those of the threshold current (I_{TH}), and thus the results of AND and OR operations could not be clearly shown. We assumed the state “1” if the output current was higher than the threshold current, and vice versa. However, after completing the training several times (blue and red solid lines), this neural network clearly determined the output states via the differences between the output current and the I_{TH} values, as shown in **Figure 4d**. In summary, the neural network was experimentally configured with the characteristics of the 2×3 synapses, and then we trained the network to be able to estimate the results of the Boolean logic operations.

[R9] Sun, Z., Ambrosi, E., Bricalli, A., Ielmini, D. Logic computing with stateful neural networks of resistive switches. *Advanced Materials* **30**, 1802554 (2018).

[R10] Zhao, S. et al. Electroluminescent synaptic devices with logic functions. *Nano Energy* **54**, 383–389 (2018).

[R11] Fuller, E. J. et al. Parallel programming of an ionic floating-gate memory array for scalable neuromorphic computing. *Science* **364**, 570–574 (2019).

We modified the manuscript as below.

In the manuscript:

“Finally, to confirm the applicability of the developed device to HW-NNs, we prepared a small-scale synapse array with a size of 2×3 and trained Boolean logic operations such as binary AND and OR onto the neural network based on the synapse array. Various logic gate operations can be implemented on this neural network platform via the training of the gate functions, as in the case of reconfigurable circuits^{1, 43, 46}.”

(5) Given the large-scale deployment of neuromorphic computing in the future, it may be a good choice to take advantage of the mature silicon processing. The authors may state the implication of the current work for neuromorphic computing based on silicon where appropriate.

ANS: The aim of this study is to demonstrate a cross-point synapse array consisting of field effect transistor (FET)-type synapses with a separated weight-control terminal. This three-terminal synapse array can be achieved by (i) adopting the vertical gate-all-around field effect transistor (GAA-FET) concept and (ii) securing acceptable gate controllability with the assistance of an ion-gel dielectric.

Notably, the GAA-FET concept has already been considered for 3-nm technology node (for a lateral type) and the next technology node (for a vertical type) by many global semiconductor companies. Thus, this research is very meaningful as a proof-of-concept of a cross-point FET-type synapse array that can be used to implement a neural network (NN) based on Si CMOS technology.

We discussed the implication of the current work for the implementation of NNs based on Si CMOS technology in the manuscript as below.

In the manuscript:

“In this study, we successfully implemented a novel crossbar synapse array based on a vertical organic transistor with an ion-gel WC layer. This three-terminal synapse array was achieved by (i) adopting the vertical gate-all-around field effect transistor (GAA-FET) concept and (ii) securing acceptable gate controllability with the assistance of an ion-gel dielectric. Mobile ions in the ion gel penetrated the free volume in the P3HT vertical organic channel located at every cross-point between the top and bottom electrode lines, which resulted in a nonvolatile change in the channel conductance.

...

A very high recognition rate of 92.5% for MNIST digit patterns was achieved in a simulated two-layer ANN with a size of $400 \times 200 \times 10$. Notably, this GAA-FET concept has already been considered for a 3-nm technology node (for a lateral type) and the next technology node (for a vertical type) by many global semiconductor companies. Thus, this research is meaningful as a proof-of-concept of a cross-point FET-type synapse array that can be used to implement NNs based on Si CMOS technology. We expect the proposed vertical crossbar synapse array to play a pioneering role in the development of high-performance and high-density NNs in the future.”

Reviewer #2 (Remarks to the Author):

The authors present a novel study ranging from vertical organic synaptic device to artificial neural network. The proposed vertical ion-gel-gated synapse could achieve a remote weight update via gate driven ion penetration in P3HT channel, allowing a 3D crossbar array based on three-terminal synaptic cells. The authors provide a generalized approach to significantly improve packing density of three-terminal synaptic devices. Additionally, the ion-gel-gated synapse show intriguing device properties including low nonlinearity (<1), large dynamic range (>10), high endurance and reproducibility. This study is innovative in device architecture, and clear in its theme. Despite I am positively impressed by structural innovation and experiment result, several critical points must be addressed to enhance the manuscript for publishing in high-impact Nature Communications.

ANS: Thank you for reviewing our paper. We appreciate your insightful comments on our research. We have revised the manuscript according to your suggestions and believe that these revisions have improved the paper.

Please find below our responses (in blue) to each of your specific comments (in black). Revisions to the original article are indicated in red.

(1) The authors claim that the conductance control mechanism is based on gate driven ion transfer between ion gel and P3HT, while mechanism research and experimental verification are missing.

ANS: As per the reviewer's suggestion, we experimentally verified the penetration of negative ions into the P3HT layer using scanning electron microscope-energy dispersive X-ray spectroscopy (SEM-EDS). A multilayered structure of P3HT/ion-gel/Au was fabricated on a highly *p*-doped Si wafer. We then applied a V_{WC} of -3 V for 5 s to the Au electrode to induce the penetration of anions (TFSI⁻) into the P3HT film. Finally, after removing the ion-gel/Au layer through a physical peel-off method, the cross section of the P3HT layer was investigated *via* SEM-EDS, as shown in **Figure R1a-c**. Elemental signals for fluorine were detected to a depth of 100 nm from the P3HT surface, indicating that the TFSI⁻ mobile ions penetrated well into the P3HT layer under a negative V_{WC} .

Figure R1. Cross-sectional SEM-EDS images of 200-nm-thick P3HT film doped with negative ions. (a) Cross-sectional SEM image of P3HT film. Mapping images of elemental signals detected from (b) carbon in polymer chain of P3HT and (c) fluorine of TFSI⁻ ions. Scale bar is 200 nm.

We added the relevant information to the manuscript and Supporting Information as below.

In the manuscript:

“By virtue of the free volume in the semiconducting polymer layer, mobile negative ions in the ion gel could readily penetrate the channel under the application of a negative WC voltage (V_{WC})^{37, 38}. **The penetration of negative ions (TFSI⁻ ions) was proven using scanning electron microscope-energy dispersive X-ray spectroscopy (SEM-EDS) analysis (see details in **Supplementary Figure 1**).** In contrast, ions moved out from the channel layer under the application of a positive V_{WC} .”

In Supporting Information:

“

Supplementary Figure 1 Cross-sectional SEM-EDS images of 200-nm-thick P3HT film doped with negative ions **a**, Cross-sectional SEM image of P3HT film. **b-c**, Mapping images of elemental signals detected from (b) carbon in polymer chain of P3HT and (c) fluorine of TFSI⁻ ions. Scale bar is 200 nm.

The movement of anions (TFSI⁻ ions) under a negative V_{WC} was observed using scanning electron microscope-energy dispersive X-ray spectroscopy (SEM-EDS). A multilayered structure of P3HT/ion-gel/Au was fabricated on a highly *p*-doped Si wafer. We then applied a V_{WC} of -3 V for 5 s to the Au electrode to induce the penetration of anions (TFSI⁻) into the P3HT film. Finally, after removing the ion-gel/Au layer through a physical peel-off method, the cross section of the P3HT layer was investigated *via* SEM-EDS, as shown in **Supplementary Figure 1a–c**. Elemental signals for fluorine were detected to a depth of 100 nm from the P3HT surface, indicating that the TFSI⁻ mobile ions penetrated well into the P3HT layer under a negative V_{WC} .”

(2) As indicated in Figure 1d, more significant conductance changes could be induced by negative V_{WC} . Why? The authors should comment. LTP/LTD characteristics were achieved by 100 consecutive potentiation pulses ($V_{WC} = -3$ V) followed by 100 consecutive depression pulses ($V_{WC} = +2$ V) in Figure 1f; while, in Figure 2 and Figure 3, the authors used V_{WC} pulses with amplitudes of ± 3 V. Please comment.

ANS: As the reviewer mentioned about the previous **Figure 1d**, the conductance change (ΔPSC) induced by a positive V_{WC} (depression) was smaller than that by a negative V_{WC} (potentiation). This was because only a few anions (TFSI⁻) penetrated the P3HT channel region under the initial condition.

To minimize this large difference between ΔPSC s by positive and negative V_{WC} s, we applied a few negative pulses to the weight-control terminal in advance, and then measured ΔPSC (EPSC and IPSC) again (**Figure R2**).

Figure R2. EPSC and IPSC responses induced by negative and positive V_{WC} s with magnitudes varying from ± 0.5 to ± 3 V.

Fig. 1d in the manuscript was revised as below.

In the manuscript:

“Then, the synaptic properties, including the EPSC/IPSC, paired-pulse facilitation (PPF), and LTP/D, of the ion-gel-gated vertical synapse were analyzed. Under varied V_{WC} with its amplitude from ± 0.5 to ± 3 V and

width of 50 ms, the PSC measured at a constant presynaptic voltage (V_{pre}) of -0.01 V showed clear EPSC and IPSC responses (Fig. 1d). The PSC immediately increased (decreased) upon the application of negative (positive) V_{WC} , and it was retained even after 50 s; however, it did not return to the initial value, because of the residual ions in the P3HT layer.

Fig. 1. d, EPSC and IPSC responses induced by negative and positive V_{WC} s with magnitudes varying from ± 0.5 to ± 3 V.”

The LTP/D characteristic shown in Figure 1f was achieved under the best pulse condition of -3 V/ $+2$ V. On the contrary, in Figure 2 and Figure 3, to clearly compare the LTP/D characteristics with respect to dimension and cycle variations, we intentionally chose a V_{WC} of $+3$ V instead of $+2$ V. This is because the depression pulses of $+3$ V were high enough to pull the anions in the P3HT channel into the ion-gel region regardless of dimension or cycle. Since this made the device return to its initial state, it was possible to investigate the LTP/D characteristics for each variation more distinctly.

To avoid confusion, because the depression pulses of $+2$ V were also high enough to pull the ions from the P3HT, we chose -3 V/ $+2$ V for potentiation/depression pulses in all measurements and investigated the LTP/D characteristics again.

The revised Figure 2 and 3 and relevant information were added to the manuscript as below.

In the manuscript:

“Fig. 2a and Supplementary Figure 3 show the LTP/D characteristics of vertical synaptic devices with P3HT channels of various thicknesses under the application of a set of V_{WC} pulses consisting of 100 potentiation pulses ($V_{WC} = -3$ V) and 100 depression pulses ($V_{WC} = +2$ V). Here, the pulse frequency and pulse width were fixed at 2 Hz and 50 ms, respectively (see Supplementary Figures 4 and 5 for the additional information about various pulse frequencies and pulse widths). The thickness of the P3HT channel was controlled in the range of 20–95 nm by varying the concentration of the P3HT solution, whereas the channel area was fixed at $50 \times 50 \mu\text{m}^2$. The maximum conductance value (G_{max}) was the highest (18.9 mS) in the device with the thinnest channel (20 nm), and it decreased to 2.3 μS with an increase in the thickness of the P3HT channel to 95 nm. Because the channel thickness is considered as the channel length in a vertically stacked device, in this study, the channel length of the synaptic device increased with increasing thickness of the P3HT channel; this resulted in a decrease in the overall channel conductance. The dynamic range (G_{max}/G_{min}) values of the synaptic devices having 20-, 35-, and 55-nm-thick P3HT channels were higher than 10, which is the minimum value required for a successful pattern recognition task⁴⁵. To extract the values of NL and NS_{eff} , which represent the precision of the weight-update behavior, we first normalized the LTP/D characteristic curves of the devices with different channel lengths by dividing each conductance value by the maximum value (G/G_{max}), as shown in Fig. 2b. Then, we calculated the NL value by fitting the measured curve to the normalized one (see detailed equations in the METHODS section and Supplementary Figure 6). The synaptic device with the thinnest channel (20 nm) showed a positive NL value (+5.4). In this device, the ions were able to penetrate the entire channel upon the application of V_{WC} pulses, which enhanced the overall channel controllability. In contrast, the NL values of the devices with the thicker channels decreased in the negative direction and reached -0.6 for the device with the 55-nm-thick

P3HT channel. This is because the channel region far away from the ion gel was relatively impermeable to the mobile ions in the thick-channel device, which degraded the controllability of channel conductance under the same number of V_{WC} pulses. Note that a similar behavior of the NL value was seen in the LTD region. The synaptic device with the 20-nm-thick channel exhibited a sharp decrease in the NL value as high as -8.7 , and the 55-nm-thick P3HT exhibited a relatively linear decreasing characteristic ($NL_{LTD} = -2.3$). **Fig. 2c** and **2d** show the absolute NL and NS_{eff} values plotted as functions of the P3HT thickness for the LTP and LTD characteristics, respectively. Here, states having ΔG above the noise range (0.5% of $G_{max} - G_{min}$) were defined as the effective states. The $|NL|$ value (denoted by black circles) was lowest (0.6/2.3 for LTP/D regions) for the device with the 55-nm-thick P3HT channel. This device also showed the highest NS_{eff} of 96/72 for the LTP/D regions.

Next, the effect of the channel area on the LTP/D characteristics was investigated by varying the width of metal lines (**Fig. 2e and 2f**). The channel areas were controlled to 30×30 , 50×50 , 70×70 , and $90 \times 90 \mu m^2$, where the thickness of the P3HT channel (that is, channel length) was fixed at 55 nm. Among these devices, the one with the largest channel area ($90 \times 90 \mu m^2$) showed the highest conductance value in the absence of any external voltage stimulus ($V_{WC} = 0$ V), which indicates that the device had the lowest channel resistance (**Supplementary Figure 7**). However, this device with the $90 \times 90 \mu m^2$ channel showed a poor G_{max}/G_{min} of 1.9, which was attributed to the obstruction of ion penetration by the large metal coverage (**Supplementary Figure 8**). The G_{max}/G_{min} value increased from 1.9 to 18 as the channel area decreased from $90 \times 90 \mu m^2$ to $30 \times 30 \mu m^2$, because of the enhanced channel controllability. The highest G_{max} and G_{min} values were obtained for the device with the $30 \times 30 \mu m^2$ channel. The absolute value of NL and the NS_{eff} value for the LTP/D regions were plotted as a function of the channel area, as shown in **Fig. 2g** and **2h**. The device with a channel area of $50 \times 50 \mu m^2$ exhibited desirable synaptic properties in both the LTP and LTD regions (i.e., low $|NL|$ values and the highest NS_{eff}). The device with the channel area of $30 \times 30 \mu m^2$ also showed a low $|NL|$ of 0.35 and high NS_{eff} of 96. However, this device had a high $|NL|$ value of 4.2 and a low NS_{eff} of 50 in the LTD region. Overall, the LTP/D characteristics of the ion-gel-gated vertical synapse were strongly affected by the ion penetration into the organic channel. From these results, the device with the channel area of $50 \times 50 \mu m^2$ and channel thickness of 55 nm was confirmed to exhibit desirable LTP/D characteristics such as large G_{max}/G_{min} , low $|NL|$, and sufficient NS_{eff} .

Fig. 2 Optimization of vertical synaptic device geometry **a**, LTP/D characteristics of thickness-controlled vertical crossbar synaptic devices under application of 100/100 potentiation/depression pulses (V_{WC} pulses with amplitudes of -3 V/ $+2$ V). Devices had P3HT channels with thicknesses of 20, 35, 50, and 70 nm. **b**, Normalized LTP/D curves of thickness-controlled synaptic devices. **c**, Plots of $|NL|$ and NS_{eff} as functions of thickness of P3HT channel in LTP region and (d) LTD region. **e**, LTP/D characteristics of synaptic devices with various areas of line cross-point. Channel area (A_{ch}) was controlled to 30×30 , 50×50 , 70×70 , and $90 \times 90 \mu m^2$ as shown in the inset. **f**, Normalized LTP/D curves of area-controlled synaptic devices. Plots of $|NL|$ and NS_{eff} as functions of area of line cross-point in **g**, LTP and **h**, LTD region. The average $|NL|$ and NS_{eff} are obtained from 5 devices prepared independently, and the error bars represents the standard deviation of

the data.

Fig. 3 Operational stability of vertical synaptic device **a**, LTP/D characteristics of vertical crossbar synapse array under application of various potentiation/depression pulse sets. **b**, Plots of $|NL|$ and NS_{eff} as functions of pulse number for single cycle. **c**, LTP/D characteristics of vertical synaptic device over 50 cycles. Number of potentiation/depression pulses for single cycle was set to 100/100. **d**, Cycle-to-cycle variations of LTP/D curve for 50 cycles. **e**, Plots of $G_{\text{max}}/G_{\text{min}}$, $|NL|$, and NS_{eff} for 50 LTP/D cycles. **f**, State stability under application of random combinations of potentiation/depression pulses with amplitudes of -3 V/ $+2$ V (left panel) and overlapping of PSC plots in first and last cycles of regular (PPD) and irregular (PPDP) pulse sets (middle and right panels).”

(3) Symmetrically programmable conductance states are desirable for facilitating ‘blind’ synaptic weight update during learning. Symmetry of conductance tuning remains to be improved in this study. The revelation of the physical mechanism (point 1) may be helpful for further device optimization.

ANS: To further investigate the symmetry of the LTP/D characteristic curves, we applied depression pulses of $+1$, $+2$, and $+3$ V to the weight-control terminal and then extracted the nonlinearities for the LTP/D regions (**Figure R3**). The potentiation pulse was fixed at -3 V. As a result, the LTP/D characteristic curve under a depression pulse of $+2$ V exhibited the best symmetry. By contrast, a depression pulse of $+1$ V was not high enough to pull the TFSI⁻ ions from the P3HT channel, and thus the PSC did not recover to the initial state. A V_{WC} of $+3$ V was sufficiently high to return the PSC level to its initial level quickly, consequently showing very asymmetric LTP/D characteristics.

Figure R3. (a) LTP/D characteristic curves measured under depression pulses of +1, +2, and +3 V, where potentiation pulse was fixed at -3 V. (b) Nonlinearities ($|NL|$) extracted from LTP/D characteristic curves.

We added the relevant results for the symmetry of the LTP/D characteristics to the Supporting Information as below.

In the manuscript:

“To investigate the repeatability and stability of the LTP/D characteristics in each cycle, various pulse sets with different numbers of pulses were applied to the WC terminal. The potentiation and depression pulses were set to -3 V and $+2$ V, respectively (see the LTD optimization procedure in **Supplementary Fig. 9**). **Fig. 3a** shows the PSC response of the synaptic device over five cycles under the application of different numbers of pulses, ranging from 5 to 100 (a total of 2000 pulses).”

In Supporting Information:

“

Supplementary Figure 9 LTD optimization of vertical synapse a, LTP/D characteristic curves measured under depression pulses of +1, +2, and +3 V, where potentiation pulse was fixed at -3 V. **b**, $|NL|$ extracted from LTP/D characteristic curves.

To further investigate the symmetry of the LTP/D characteristic curves, we applied depression pulses of +1, +2, and +3 V to the weight-control terminal, and then extracted the $|NL|$ s for the LTP/D regions (**Supplementary Figure 9**). The potentiation pulse was fixed at -3 V. As a result, the LTP/D characteristic curve under a depression pulse of +2 V exhibited the best symmetry. By contrast, a depression pulse of +1 V was not high enough to pull the TFSI ions from the P3HT channel, and thus the PSC did not recover to the initial state. A V_{WC} of +3 V was sufficiently high to return the PSC level to its initial level quickly, consequently showing very asymmetric LTP/D characteristics.”

(4) The font (red) in Fig. 1c is too small to read. Please modify.

ANS: We increased the size of the font.

(5) The reviewer recommends the authors to include “Adv. Mater. 2018, 30, 1802883; Adv. Funct. Mater. 2019, 29, 1902374” as references for “three-terminal artificial synapses implemented with various inorganic and organic materials.....” (Page 2, line 51).

ANS: Per the reviewer’s suggestion, we added the recommended references to the References section as below.

In the References:

“34. Lv, Z. Mimicking neuroplasticity in a hybrid biopolymer transistor by dual modes modulation. *Adv. Funct. Mater.* **29**, 1902374 (2019).

35. Wang, Y. et. Al. Photonic synapses based on inorganic perovskite quantum dots for neuromorphic computing. *Adv. Mater.* **30**, 1802883 (2018).”

In the manuscript:

“In recent studies, three-terminal artificial synapses implemented with various inorganic and organic materials showed a desirable weight-controllability property *via* various charge-storage principles using interfacial traps²⁸⁻³⁰, atomic vacancies¹⁴, ion intercalation^{22, 26, 28, 31}, and floating gates³²⁻³⁵.”

(6) What does WC stand for? The authors should give the full name of the abbreviation.

ANS: “WC” stands for “weight-control.” We corrected it in the manuscript as below.

In the manuscript:

“For the device configuration, a sub-100-nm-thick poly(3-hexylthiophene) (P3HT) channel is positioned at every cross-point of the pre- and postsynaptic terminals, and the ion-gel **weight-control (WC)** layer is deposited on them.

...

The ion-gel layer and top gate lines were utilized to achieve the nonvolatile-weight-change property of a biological synapse as a **weight-control (WC)** terminal stack.”

(7) Fixing device variability is a prerequisite for mass production. What about the device-to-device variability?

ANS: We already discussed the device-to-device variation with 10 synaptic devices in terms of the recognition rate in the manuscript (see **Figure 4f** and **4g**). We fabricated 10 synaptic devices and then extracted the (i) maximum and minimum current values (I_{\max} , I_{\min}) and (ii) nonlinearities (NLs) from the LTP/LTD characteristic curves of the devices (see the extracted parameters in **Supplementary Table 2**). Using these parameters, we performed training/recognition tasks for the MNIST digit patterns and estimated the recognition rates at every 40,000 training numbers (1 epoch) for every device. As a result, the standard deviation of the maximum recognition rates was as low as 2.5%, and the standard deviation was 4.2% even after 25-epoch learning.

Figure 4. (g) Maximum (blue) and final (black) recognition rates of 10 synaptic devices.

Reviewer #3 (Remarks to the Author):

The paper presents a type of ionic transistor, amenable to 3d integration, and demonstrates the use in training neural networks. The device structure's compatibility with efficient 3d integration, combined with enhanced analog capabilities of an ionic device are the main novel features claimed in the work. This device structure is novel, and the work is original, and this is a topic of significant recent interest. However, the characteristics are not competitive with that found in top state of the art published results. I do not find the results impactful enough to recommend publication in Nature Communications. A future version of this work might be suitable for publication, if a number of issues are addressed:

ANS: Thank you for reviewing our paper. We appreciate your insightful comments on our research. We have revised the manuscript according to your suggestions and believe that these revisions have improved the paper.

Please find below our responses (in blue) to each of your specific comments (in black). Revisions to the original article are indicated in red.

(1) 92.5% accuracy on MNIST training does not represent state of the art. In state of the art devices, the full precision matches the ionic device training accuracy which is about 98% (see ref 1). Also, the conductance versus pulse width curves show nice linearity for many of the increasing conductance cases/LTP, but all curves show a very sharp nonlinear drop for decreasing conductance/LTD. The plots of NL in Fig. 2c and 2f appear only to take into account the increasing cases/LTP. In our experience, highly nonlinear LTD coupled with asymmetric LTP versus LTD precludes reasonable accuracy for training even simple datasets like MNIST. Was this LTD nonlinearity included in the neurosim modeling and still a 92.5% accuracy in Fig. 4? If this accuracy was obtained even with nonlinear decrease, then the authors should discuss if a scheme was used to mitigate asymmetric nonlinearity (because it is surprising that neurosim would give even this moderately high MNIST accuracy with that level of LTD NL/asymmetry without a mitigation scheme).

ANS: To investigate the feasibility of the proposed crossbar array concept toward hardware neural networks (HW-NNs), we performed training/recognition tasks for MNIST digit patterns. The simulation was conducted using the platform “MLP+NeuroSim ver. 1.0.” The simulated network consisted of 400 input neurons, 200 hidden neurons, and 10 output neurons. For the simulation, we used the following nonideal device properties: i) maximum and minimum currents (I_{\max} , I_{\min}) and ii) nonlinearities (NLs) extracted in the LTP/D characteristic curves from 10 synaptic devices (other parameters remained ideal). Among the cases simulated with the 10 synaptic devices, a maximum accuracy of 92.5% was obtained when the NL values for LTP/D and the dynamic range were $-1.25/-5.72$ and 10.72 , respectively (**Figure 4f**). An accuracy of 92.5% is meaningful when compared to the 96% accuracy of the ideal case.

To further enhance the accuracy of the MNIST training, NL in the LTD region needed to be improved. For this, we reduced the amplitude of the depression pulses from +3 V to +2 V and then investigated the LTP/D characteristics again. As shown in **Figure R1a**, NL of LTD was improved to -2.27 . Furthermore, owing to this improvement, a maximum accuracy of 94% could be achieved (**Figure R1b**). Here, the size of the simulated NN was $400 \times 100 \times 10$, and its maximum accuracy (ideal case) was 96.2%.

Figure R1. (a) LTP/D characteristic curve measured under $V_{LTP/D} = -3\text{ V}/+2\text{ V}$, where $NL_{LTP/D} = -0.63/-2.27$. (b) Recognition rate vs. training epoch of best linearity case (black) and ideal case (blue).

We briefly mentioned the further improvement in the accuracy and added the improved LTP/D characteristic curves and the recognition rate of the best linearity case to the Supporting Information as below.

In the manuscript:

“The minimum accuracy was 85.7% when the NL value for LTP/D and the dynamic range were $-0.42/-6.77$ and 49.33, respectively. Further improvement in the accuracy was achieved later by pulse engineering (Supplementary Figure 13). We then investigated the device-to-device variation in the recognition rate for the 10 synaptic devices.”

In Supporting Information:

“

Supplementary Figure 13 MNIST application of vertical synapse a, LTP/D characteristic curve measured under $V_{LTP/D} = -3\text{ V}/+2\text{ V}$, where $NL_{LTP/D} = -0.63/-2.27$. **b**, Recognition rate vs. training epoch of best linearity case (black) and ideal case (blue).

To further enhance the accuracy of the MNIST training, NL in the LTD region needed to be improved. For this, we reduced the amplitude of the depression pulses from $+3\text{ V}$ to $+2\text{ V}$ and investigated the LTP/D characteristics again. As shown in Supplementary Figure 13a, NL of LTD was improved to -2.27 . Furthermore, owing to this improvement, a maximum accuracy of 94% could be achieved (Supplementary Figure 13b). Here, the size of the simulated NN was $400 \times 100 \times 10$, and its maximum accuracy (ideal case) was 96.2%.”

(2) The STP investigation appears to show a state decay on the order of seconds. Although this is useful for STP, it is a problem for neural network training. If the weights are quickly decaying significantly during the time of a few training epochs, this will degrade the accuracy on benchmark training, such as MNIST. Please discuss if and how this drift was accounted for in the theoretical study of the MNIST training described on pg 8 and Fig. 4.

ANS: In the short-term plasticity (STP) investigation (**Figure 1e**), we intentionally chose a small weight-control voltage (V_{WC}) of -1 V to find the postsynaptic current (PSC) response according to the paired-pulse interval. Because a V_{WC} of -1 V was insufficient to retain the penetrated ions inside the P3HT channel, the PSC quickly decayed on the order of seconds, showing no long-term memory property.

On the contrary, to analyze the long-term plasticity, we applied larger potentiation/depression voltages of -3 V/ $+2$ V to the weight-control terminal (see EPSC/IPSC characteristics in the previous **Figure 1d**). However, the PSCs for potentiation and depression pulses were slightly decayed. To investigate the state decaying effects on the neural network training, we measured the LTP/D characteristics under pulse frequencies of 2, 4, 6, and 10 Hz, and then conducted the training/recognition tasks for MNIST patterns. The pulse amplitudes were fixed at -3 V/ $+2$ V for potentiation/depression, respectively. As the pulse frequency varied from 2 to 10 Hz, the dynamic range increased from 20.9 to 39.6, and the linearity degraded from 0.72/ -5.04 to 0.78/ -5.2 , respectively (**Figure R2a**). Owing to the trade-off relationship between the dynamic range and linearity with respect to the pulse frequency, the recognition rate stayed in the range of 90 to 91% (**Figure R2b**).

Figure R2. (a) LTP/D characteristic curves measured under pulse frequencies of 2, 4, 6, and 10 Hz, where $V_{LTP/D} = -3$ V/ $+2$ V. (b) Recognition rates with respect to pulse frequency.

We mentioned additional information about various pulse frequencies in the manuscript and added the experimental results and simulation data to the Supporting Information as below.

In the manuscript:

“**Fig. 2a** and **Supplementary Figure 3** show the LTP/D characteristics of vertical synaptic devices with P3HT channels of various thicknesses under the application of a set of V_{WC} pulses consisting of 100 potentiation pulses ($V_{WC} = -3$ V) and 100 depression pulses ($V_{WC} = +2$ V). Here, the pulse frequency and pulse width were fixed at 2 Hz and 50 ms, respectively (see **Supplementary Figures 4** and **5** for the additional information about various pulse frequencies and pulse widths). The thickness of the P3HT channel was controlled in the range of 20–95 nm by varying the concentration of the P3HT solution, whereas the channel area was fixed at $50 \times 50 \mu\text{m}^2$.”

In Supporting Information:

“

Supplementary Figure 4 Thickness-dependent electrical property of vertical synapse a, LTP/D characteristic curves measured under pulse frequencies of 2, 4, 6, and 10 Hz, where $V_{LTP/D} = -3 \text{ V}/+2 \text{ V}$. b, Recognition rates with respect to pulse frequency.

To investigate the state decaying effects on the neural network training, we measured the LTP/D characteristics under pulse frequencies of 2, 4, 6, and 10 Hz, and then conducted the training/recognition tasks for MNIST patterns. The pulse amplitudes were fixed at $-3 \text{ V}/+2 \text{ V}$ for potentiation/depression, respectively. As the pulse frequency varied from 2 to 10 Hz, the dynamic range increased from 20.9 to 39.6, and the linearity degraded from 0.72/ -5.04 to 0.78/ -5.2 , respectively (**Supplementary Figure 4a**). Owing to the trade-off relationship between the dynamic range and linearity with respect to the pulse frequency, the recognition rate stayed in the range of 90 to 91% (**Supplementary Figure 4b**)."

(3) Writing the device requires a pulse width of 50ms. This is far too slow to be useful for the VMM-type applications of interest (for training a deep network), such as those discussed in references 1, 9, and 18. Related work has been able to show 200ns programming (i.e. reference 1). Is there a path, perhaps through scaling for the proposed system to achieve sub-microsecond write speeds?

ANS: In this work, we intentionally used a slow weight-control voltage (V_{WC}) with a 50-ms pulse width to induce sufficient ion penetration of the P3HT channel region. However, we agree with the reviewer's opinion that a high-speed write operation should be considered for training complex hardware neural networks.

To investigate the sub-microsecond writing capability, we applied a V_{WC} s with three pulse widths of 300, 500, and 700 ns to the weight-control terminal, and then analyzed the postsynaptic current (PSC) responses (**Figure R3**). The pulse amplitudes were fixed at $-3 \text{ V}/+2 \text{ V}$ for potentiation/depression, respectively. As shown in **Figure R3a**, our device exhibited clear weight-updating behaviors of potentiation/depression for all cases of the pulse widths.

The LTP/D characteristics showed a relatively small dynamic range and high nonlinearity when compared to the values for the previous slow-speed write operation (**Figure R3b**). It seems that only a few ions penetrated the P3HT channel region owing to their slow speed under high-speed operations, consequently degrading the channel controllability. Therefore, we believe that device scaling is a good solution for improving the channel controllability under sub-microsecond write/read operations.

Figure R3. (a) Postsynaptic current (PSC) responses with respect to potentiation (upper panel, $V_{\text{WC}} = -3 \text{ V}$) and depression (lower panel, $V_{\text{WC}} = +2 \text{ V}$) voltage pulses with pulse widths of 300, 500, and 700 ns. (b) LTP/D characteristics measured under application of 100/100 potentiation/depression pulses ($V_{\text{WC}} = -3 \text{ V}/+2 \text{ V}$) with pulse widths of 300, 500, and 700 ns.

We stated additional information about various pulse widths in the manuscript and added the experimental results and discussions about the sub-microsecond write operation of the proposed device to the Supporting Information as below.

In the manuscript:

“**Fig. 2a** and **Supplementary Figure 3** show the LTP/D characteristics of vertical synaptic devices with P3HT channels of various thicknesses under the application of a set of V_{WC} pulses consisting of 100 potentiation pulses ($V_{\text{WC}} = -3 \text{ V}$) and 100 depression pulses ($V_{\text{WC}} = +2 \text{ V}$). Here, the pulse frequency and pulse width were fixed at 2 Hz and 50 ms, respectively (see **Supplementary Figures 4** and **5** for the additional information about various pulse frequencies and pulse widths). The thickness of the P3HT channel was controlled in the range of 20–95 nm by varying the concentration of the P3HT solution, whereas the channel area was fixed at $50 \times 50 \mu\text{m}^2$.”

In Supporting Information:

“

Supplementary Figure 5 Pulse width-dependent electrical property of vertical synapse a, PSC responses with respect to potentiation (upper panel, $V_{\text{WC}} = -3 \text{ V}$) and depression (lower panel, $V_{\text{WC}} = +2 \text{ V}$) voltage pulses with pulse widths of 300, 500, and 700 ns. **b**, LTP/D characteristics measured under application of 100/100 potentiation/depression pulses ($V_{\text{WC}} = -3 \text{ V}/+2 \text{ V}$) with pulse widths of 300, 500, and 700 ns.

To investigate the sub-microsecond writing capability, we applied a V_{WC} with three pulse widths of 300, 500, and 700 ns to the weight-control terminal, and then analyzed the PSC responses (**Supplementary Figure 5**). The pulse amplitudes were fixed at $-3 \text{ V}/+2 \text{ V}$ for potentiation/depression, respectively. As

shown in **Supplementary Figure 5a**, our device exhibited clear weight-updating behaviors of potentiation/depression for all cases of the pulse widths.

The LTP/D characteristics showed a relatively small dynamic range and high nonlinearity when compared to the values for the previous slow-speed write operation (**Supplementary Figure 5b**). It seems that only a few ions penetrated the P3HT channel region owing to their slow speed under high-speed operations, consequently degrading the channel controllability. Therefore, we believe that device scaling is a good solution for improving the channel controllability under sub-microsecond write/read operations.”

(4) Is 10k pulses the maximum number of cycles before degradation occurs? Is there a reason not to measure out to a greater number of cycles? It seems more than 10k pulses will be needed even to train MNIST, and if there is degraded analog performance later in the training process this should be accounted for in the model.

ANS: For our devices, the maximum number of cycles where the degradation occurs is not 10k pulses. To clarify this issue, we fabricated new devices and then extracted the nonlinearities (NL s) and dynamic range (G_{\max}/G_{\min}) values at every 40k-th pulse from the LTP/D characteristics measured further to 200k pulses (**Figure R4a** and **R4b**). As a result, the $|NL|$ values were almost unchanged for 200k pulses, and G_{\max}/G_{\min} degraded below 10 after applying 120k pulses. The corresponding simulation results showed that the training/recognition performance was maintained perfectly up to 80k pulses (**Figure R4c**). This is probably owing to the stability of the $|NL|$ values despite the degradation of G_{\max}/G_{\min} .

Figure R4. (a) Postsynaptic current (PSC) responses measured further to 200k pulses. (b) Nonlinearity ($|NL|$) and dynamic range (G_{\max}/G_{\min}) extracted at every 40k-th pulse from LTP/D characteristic curves. (c) Recognition rate vs. number of pulses for artificial neural networks composed of parameters from (b).

We added the relevant information to the manuscript and Supporting Information as below.

In the manuscript:

“The cycle-to-cycle variations of $|NL|$, NS_{eff} , and G_{\max}/G_{\min} were calculated to be 1.4%, 6.4%, and 1.1%, respectively. Additionally, we investigated the LTP/D characteristics by elongating the cycle test to 200k pulses for observing the degradation of the device performance (**Supplementary Figure 11**). Then, reliability of the PSC was also investigated under irregular-pulse conditions. **Fig. 3f** shows a plot of the real-time change in the PSC of the device under the application of two different V_{WC} pulse sets.”

In Supporting Information:

“

Supplementary Figure 11 Long-term stability of vertical synapse **a**, PSC responses measured further to 200k pulses. **b**, $|NL|$ and G_{\max}/G_{\min} extracted at every 40k-th pulse from LTP/D characteristic curves. **c**, Recognition rate vs. number of pulses for artificial neural networks composed of parameters from (b).

We fabricated new devices and then extracted the NL s and G_{\max}/G_{\min} values at every 40k-th pulse from the LTP/D characteristics measured further to 200k pulses (**Supplementary Figure 11a** and **11b**). As a result, the $|NL|$ values were almost unchanged for 200k pulses, and G_{\max}/G_{\min} degraded below 10 after applying 120k pulses. The corresponding simulation results showed that the training/recognition performance was maintained perfectly up to 80k pulses (**Supplementary Figure 11**). This is probably owing to the stability of the $|NL|$ values despite the degradation of G_{\max}/G_{\min} .

(5) Write currents and energies need to be discussed. As discussed in many of this article’s references, low energy DNN training is a key advantage for analog, and a key reason we are interested in these new devices. With the long write time, if the device requires even 1uA, the write energy will be on the order of 50ms x 1uA x 3V = 150nJ. This is several orders of magnitude higher than the energy of <10pJ published in recent works. If the write energy is in fact very high, a path to scaling and significantly reducing it should be discussed as well.

ANS: As the reviewer suggested, we roughly calculated a writing energy using the equation below:

$$E_{\text{writing}} = V_{\text{amp}} \times I_{\text{peak}} \times t_{\text{WC}} \quad (1)$$

V_{amp} and I_{peak} represent the amplitude of the weight-control voltage (V_{WC}) and the peak value of the writing current, respectively. t_{WC} is the pulse width of V_{WC} . As shown in **Figure R5a**, the writing energy (E_{writing}) for the potentiation pulse was $-3 \text{ V} \times -79.1 \text{ nA} \times 50 \text{ ms} = 11.9 \text{ nJ}$, and E_{writing} for the depression pulse was $2 \text{ V} \times 15.9 \text{ nA} \times 50 \text{ ms} = 1.6 \text{ nJ}$. Such values are higher than the energies below 10 pJ reported in recent works, and thus the values must be reduced further to be applied to hardware neural networks.

One way to reduce E_{writing} is to scale down the channel area. To investigate the relationship between E_{writing} and the channel area, we experimentally extracted the E_{writing} values for the potentiation/depression pulses when the channel areas were 30^2 , 50^2 , and $70^2 \mu\text{m}^2$, as shown in **Figure R5b**. The writing energies for the potentiation/depression pulses decreased from 13.3/1.9 nJ to 4.6/0.77 nJ, respectively, as the channel area was scaled down.

Another possibility to reduce E_{writing} was confirmed through pulse width engineering (**Figure R5c**). When the pulse width was scaled down to 70 μs , the writing energies for the potentiation/depression pulses decreased to 2.6/0.3 nJ. These energies were reduced further to 0.25/0.17 nJ under a pulse width of 700 ns.

Based on these experimental results, a writing energy of tens of pJ is expected to be achieved by scaling down the channel area to sub- $5^2 \mu\text{m}^2$ and reducing the pulse width to 500 ns.

Figure R5. (a) Writing currents ($I_{writing}$) measured when applying potentiation/depression pulses to weight-control terminal. Writing energies consumed by single potentiation/depression pulses as function of (b) channel area and (c) pulse width.

We mentioned the writing energy of the device in the manuscript and added the discussion about the scaling of writing energies to the Supporting Information as below.

In the manuscript:

“Overall, the optimized vertical synaptic device showed stable weight-update behavior under various V_{WC} conditions. Additionally, we investigated the writing energy of the device for single potentiation/depression pulse by measuring the current between WC terminal and postsynaptic terminal (see details in **Supplementary Figure 12**). The device exhibited energy consumption of 11.9/1.6 nJ for the potentiation/depression pulse, which was further reduced to 0.25/0.17 nJ under a pulse width of 700 ns.”

In Supporting Information:

Supplementary Figure 12 Energy consumption of vertical synapse a, Writing currents ($I_{writing}$) measured when applying potentiation/depression pulses to weight-control terminal. **b-c**, Writing energies consumed by single potentiation/depression pulses as function of (b) channel area and (c) pulse width.

We calculated an approximate writing energy using the equation below.

$$E_{writing} = V_{amp} \times I_{peak} \times t_{WC} \quad (1)$$

where V_{amp} and I_{peak} represent the amplitude of the weight-control voltage (V_{WC}) and the peak value of the writing current, respectively. t_{WC} is the pulse width of the V_{WC} . As shown in **Supplementary Figure 12a**, the writing energy ($E_{writing}$) for the potentiation pulse was $-3 \text{ V} \times -79.1 \text{ nA} \times 50 \text{ ms} = 11.9 \text{ nJ}$, and the $E_{writing}$ for the depression pulse was $2 \text{ V} \times 15.9 \text{ nA} \times 50 \text{ ms} = 1.6 \text{ nJ}$. Such values are higher than the energies below 10 pJ published in recent works, and thereby, it is required to be reduced further to be applied for hardware neural networks.

A possible method to reduce the $E_{writing}$ is to scale down a channel area. To investigate the relationship between the $E_{writing}$ and the channel area, we experimentally extracted the $E_{writing}$ values for the potentiation/depression pulses when the channel areas were 30^2 , 50^2 , and $70^2 \mu\text{m}^2$, as shown in **Supplementary Figure 12b**. The writing energies for the potentiation/depression pulses decreased from 13.3/1.9 nJ to 4.6/0.77 nJ, respectively, the channel area was scaled down. Furthermore, another possibility

to reduce the E_{writing} was confirmed through pulse width engineering (**Supplementary Figure 12c**). When the pulse width was scaled down to 70 μs , the writing energies for the potentiation/depression pulses decreased to 2.6/0.3 nJ. These energies were reduced further to 0.25/0.17 nJ under the pulse width of 700 ns.

Based on these experimental results, the writing energy of tens of pJ is expected to be achieved by scaling down the channel area to sub- $5^2 \mu\text{m}^2$ and reducing the pulse width to 500 ns.”

(6) The 3D compatible process is one of the potential advantages of this device. However, as with other similar recently demonstrated ionic analog devices, it would still need to be integrated with CMOS to control the arrays. The authors did not comment or consider CMOS compatibility. It seems that some of the materials, such as the P3HT would probably not be able to withstand the CMOS back end of line temperatures (typically about 400C). Would this be a likely showstopper for this device?

ANS: As the reviewer mentioned, the 3D-compatible process with a low thermal budget is an advantage of this vertical synapse device technology. Another important point of this work is the demonstration of a cross-point synapse array consisting of field effect transistor (FET)-type synapses with a separated weight-control terminal. This three-terminal synapse array can be achieved by (i) adopting the vertical gate-all-around field effect transistor (GAA-FET) concept and (ii) securing an acceptable gate controllability with the assistance of an ion-gel dielectric.

Notably, the GAA-FET concept has already been considered for 3-nm technology node (for a lateral type) and next technology node (for a vertical type) by many global semiconductor companies. Thus, this research is very meaningful as a proof-of-concept of a cross-point FET-type synapse array that can be used to implement neural networks (NNs) based on Si CMOS technology. In this case, a Si vertical channel and a high-K dielectric can be considered instead of the P3HT channel and the ion-gel dielectric, respectively.

We added the implications of the current work with regard to an NN based on Si CMOS technology to the manuscript as below.

In the manuscript:

“In this study, we successfully implemented a novel crossbar synapse array based on a vertical organic transistor with an ion-gel WC layer. **This three-terminal synapse array was achieved by (i) adopting the vertical gate-all-around field effect transistor (GAA-FET) concept and (ii) securing acceptable gate controllability with the assistance of an ion-gel dielectric.** Mobile ions in the ion gel penetrated the free volume in the P3HT **vertical** organic channel located at every cross-point between the top and bottom electrode lines, which resulted in a nonvolatile change in the channel conductance.

...

A very high recognition rate of 92.5% for MNIST digit patterns was achieved in a simulated two-layer ANN with a size of $400 \times 200 \times 10$. **Notably, this GAA-FET concept has already been considered for 3-nm technology nodes (for a lateral type) and next technology node (for a vertical type) by many global semiconductor companies. Thus, this research is meaningful as a proof-of-concept of a cross-point FET-type synapse array that can be used to implement NNs based on Si CMOS technology. We expect the proposed vertical crossbar synapse array to play a pioneering role in the development of high-performance and high-density NNs in the future.**”

Reviewers' Comments:

Reviewer #1:

Remarks to the Author:

The authors have largely addressed the issues. I do not have further comments.

Reviewer #2:

Remarks to the Author:

Despite I am positively impressed by device innovation, one point (or a serious mistake) makes me quite confused and must be addressed: The data of EPSC and IPSC in manuscript and revised manuscript are quite different, under the same test condition (Figure 1d, as below). Do the authors make a mistake here? Or, the considerable data difference originates from device-to-device variations? The authors demonstrated crossbar memory array here, while, the reviewer thinks that memory cells with such a high degree of variability are not suitable for high-density integration. The same problem also exists in Figure 1f, Figure 2a, Figure 2d. For publishing in high-impact Nature Communications, this kind of mistake or device variability are unacceptable. Please comment.

Reviewer #3:

Remarks to the Author:

The authors did a commendable job addressing, in significant detail, each of the reviewer concerns I laid out in my original review. I was particularly impressed by the additional energy/scaling analysis, and I think this additional work has made a much stronger paper.

Optionally, the authors might want to add a comment on the endurance - is there a future research direction that might help maintain the higher G_{max}/G_{min} for a greater number of cycles? This should be considered optional and should not delay publication.

In summary, I recommend publication of this version of the manuscript in Nature Communications.

Point-by-Point Response to the Reviewers

Reviewer #2 (Remarks to the Author):

Despite I am positively impressed by device innovation, one point (or a serious mistake) makes me quite confused and must be addressed: The data of EPSC and IPSC in manuscript and revised manuscript are quite different, under the same test condition (Figure 1d, as below). Do the authors make a mistake here? Or, the considerable data difference originates from device-to-device variations? The authors demonstrated crossbar memory array here, while, the reviewer thinks that memory cells with such a high degree of variability are not suitable for high-density integration. The same problem also exists in Figure 1f, Figure 2a, Figure 2d. For publishing in high-impact Nature Communications, this kind of mistake or device variability are unacceptable. Please comment.

ANS: Thank you for reviewing our paper. We appreciate your insightful comments on our research. We have revised the manuscript according to your suggestions and believe that the revision has improved the paper.

During the revision process, we purchased two new bottles of P3HT from Sigma-Aldrich and it was confirmed that those products have different lot number from previously used one. The lot number of the previous bottle is MKCJ1406, while new bottles have lot number of MKCK1947. All figures in the revised manuscript were replaced with the results obtained using new P3HT. We believe that the difference in device performance observed in previous and revised versions of the manuscript is due to the different quality of P3HT rather than device-to-device variation. We have added the material information in the manuscript as follows.

In methods:

“Regioregular P3HT (M_n : 54,000–75,000, lot number: MKCK1947), PVdF-HFP (M_n : 110,000), and 1-ethyl-3-methylimidazolium bis(trifluoromethylsulfonyl)imide (EMIM:TFSI) ionic liquid were also purchased from Sigma-Aldrich.”

Reviewer #3 (Remarks to the Author):

The authors did a commendable job addressing, in significant detail, each of the reviewer concerns I laid out in my original review. I was particularly impressed by the additional energy/scaling analysis, and I think this additional work has made a much stronger paper. Optionally, the authors might want to add a comment on the endurance - is there a future research direction that might help maintain the higher G_{max}/G_{min} for a greater number of cycles? This should be considered optional and should not delay publication. In summary, I recommend publication of this version of the manuscript in Nature Communications.

ANS: Thank you for reviewing our paper. We have revised the manuscript according to your suggestion as below.

In discussion:

“A very high recognition rate of 92.5% for MNIST digit patterns was achieved in a simulated two-layer ANN with a size of $400 \times 200 \times 10$. To implement a hardware ANN with the vertical organic synapses as a follow-up research, the excellent endurance of the synapses is critically required. In this regard, identifying and understanding the failure mechanism for weight update will help in assessing and improving the endurance. Besides, the researches optimizing encapsulation layers, ion-gel dielectrics, and organic semiconductors in the synapses need to be done for the excellent endurance. Notably, this GAA-FET concept...”

Reviewers' Comments:

Reviewer #2:

Remarks to the Author:

The authors have addressed the issues. This paper can be accepted by Nature Communications now.

Reviewer #3:

Remarks to the Author:

I recommend this version for review. They have previously addressed all major concerns, and in this version they have even addressed the optional comment on endurance.